# Oligodendrocyte-encoded Kir4.1 function is required for axonal integrity

Lucas Schirmer[1,2,3,4], Wiebke Möbius[5,6], Chao Zhao[4,7], Andrés Cruz-Herranz[8,9], Lucile Ben Haim[1,2], Christian Cordano[8,9], Lawrence R Shiow[1,2], Kevin W Kelley[1,2], Boguslawa Sadowski[5,6], Garrett Timmons[8,10], Anne-Katrin Pröbstel[8,9], Jackie N Wright[1,2], Jung Hyung Sin[8,9], Michael Devereux[8,9], Daniel E Morrison[4,7], Sandra M Chang[1,2], Khalida Sabeur[1,2], Ari J Green[8,10,11], Klaus-Armin Nave[5,6], Robin JM Franklin[4,7], David H Rowitch[1,2,3,4,12]*

[1]Eli and Edythe Broad Center of Regeneration Medicine and Stem Cell Research, University of California, San Francisco, San Francisco, California, United States; [2]Department of Pediatrics, University of California, San Francisco, San Francisco, United States; [3]Department of Paediatrics, University of Cambridge, Cambridge, United Kingdom; [4]Wellcome Trust-Medical Research Council Stem Cell Institute, University of Cambridge, Cambridge, United Kingdom; [5]Department of Neurogenetics, Max Planck Institute of Experimental Medicine, Göttingen, Germany; [6]Center for Nanoscale Microscopy and Molecular Physiology of the Brain (CNMPB), Göttingen, Germany; [7]Department of Clinical Neurosciences, University of Cambridge, Cambridge, United Kingdom; [8]Department of Neurology, University of California, San Francisco, San Francisco, United States; [9]Weill Institute for Neurosciences, University of California, San Francisco, San Francisco, United States; [10]Weill Institute for Neurosciences, University of California, San Francisco, San Francisco, United States; [11]Department of Ophthalmology, University of California, San Francisco, San Francisco, United States; [12]Department of Neurosurgery, University of California, San Francisco, San Francisco, United States

*For correspondence:
dhr25@medschl.cam.ac.uk

**Abstract** Glial support is critical for normal axon function and can become dysregulated in white matter (WM) disease. In humans, loss-of-function mutations of *KCNJ10,* which encodes the inward-rectifying potassium channel KIR4.1, causes seizures and progressive neurological decline. We investigated Kir4.1 functions in oligodendrocytes (OLs) during development, adulthood and after WM injury. We observed that Kir4.1 channels localized to perinodal areas and the inner myelin tongue, suggesting roles in juxta-axonal $K^+$ removal. Conditional knockout (cKO) of OL-*Kcnj10* resulted in late onset mitochondrial damage and axonal degeneration. This was accompanied by neuronal loss and neuro-axonal dysfunction in adult OL-*Kcnj10* cKO mice as shown by delayed visual evoked potentials, inner retinal thinning and progressive motor deficits. Axon pathologies in OL-*Kcnj10* cKO were exacerbated after WM injury in the spinal cord. Our findings point towards a critical role of OL-Kir4.1 for long-term maintenance of axonal function and integrity during adulthood and after WM injury.
DOI: https://doi.org/10.7554/eLife.36428.001

## Introduction

Glial support of axons is essential for the maintenance of normal function in the central nervous system (CNS) (*Nave, 2010*; *Nave and Trapp, 2008*). For example, oligodendrocytes (OLs) maintain metabolic and trophic support of axons by providing lactate in response to sensing of axonal firing

(*Lee et al., 2012*; *Fünfschilling et al., 2012*; *Saab et al., 2016*). Exchange and buffering of ions such as $K^+$ between astrocytes (AS) and neurons have been well described and led to the current understanding that those cells are major regulators of neuronal excitability (*Cui et al., 2018*; *Tong et al., 2014*; *Kelley et al., 2018*). However, not much is known about OL-dependent regulation of axonal excitability through buffering of ions like $K^+$ during action potential propagation. Here, we focused on Kir4.1 (*Kcnj10*), a highly conserved ATP- and pH-sensitive $K^+$ channel expressed in both AS and OL cells of the CNS (*Hibino et al., 2004*; *Tanemoto et al., 2000*; *Hibino et al., 2010*). Kir channels regulate $K^+$ transmembrane gradients (*Rash, 2010*; *Menichella et al., 2006*; *Olsen and Sontheimer, 2008*; *Chever et al., 2010*), which are critical for action potential propagation as well as axonal $K^+$ outflow that is necessary to establish resting membrane potential and neuronal repolarization (*Yasuda et al., 2008*; *Seifert et al., 2009*; *Djukic et al., 2007*; *Bay and Butt, 2012*; *Sibille et al., 2015*). Its complex expression pattern, including homo- and heterotetrameric association with Kir5.1 (*Kcnj16*), underlies potentially diverse roles depending on glial cell sub-type and CNS anatomical regions (*Cui et al., 2018*; *Tong et al., 2014*; *Kelley et al., 2018*; *Hibino et al., 2004*; *Larson et al., 2018*). However, precise functions for Kir4.1 in OLs versus AS are only poorly understood.

Homotetrameric Kir4.1 is the major inward-rectifying $K^+$ channel in OLs, and its downregulation has been reported in glial cells in CNS disease (*Zurolo et al., 2012*; *Eberhardt et al., 2011*; *Schirmer et al., 2014*). Because glial Kir4.1 channels help remove extracellular $K^+$ during neuronal activity, general loss-of-function is associated with severe human neurological conditions. Human congenital *KCNJ10* loss-of-function mutations in EAST/SeSAME syndrome cause electrolyte imbalance, seizures, pathological changes in the retina, sensorineural deafness and progressive motor deficits in affected individuals (*Bockenhauer et al., 2009*; *Cross et al., 2013*; *Scholl et al., 2009*; *Thompson et al., 2011*). Prior reports have found early lethality in young adult mice lacking glial-*Kcnj10*, attributable to increased neuronal excitability and epileptic activity (*Djukic et al., 2007*; *Larson et al., 2018*). In particular, conditional ablation of *Kcnj10* using *Gfap-cre* (that targets both AS and OL glial cell types) was lethal before P30 (*Djukic et al., 2007*; *Neusch et al., 2001*), an early/ severe phenotype that precluded study of late functions in adult white matter (WM).

In contrast to prior studies, we studied the role of OL-Kir4.1 channels in long-term maintenance of axonal function during adulthood and white matter injury focusing on long white matter tracts, such as the optic nerves and the spinal cord. We found early and late roles for Kir4.1 in OL progenitor cells (OPCs) and myelinating OLs using two distinct *cre* lines to study age- and disease-related functions during development, adulthood and in the setting of WM injuries. We observed that Kir4.1 is localized to both OL cell bodies and myelin, where it is found within the inner tongue of myelin and in AS processes near the node of Ranvier, which appear poised to remove axonal $K^+$. By dissecting out early from late developmental functions, we found that OL-Kir4.1 conditional knockout (cKO) was dispensable for early myelin production but resulted in pronounced late onset axonal degeneration with damage to mitochondria in long fiber tracts of the optic nerve (ON) and spinal WM as well as after focal WM demyelination. Hence, our data suggest that $K^+$ clearance via OL-Kir4.1 channels is critical for sustained axonal function and integrity.

## Results

### OL-Kir4.1 is gradually upregulated during early postnatal development and shows a peri-axonal expression pattern

As Kir4.1 channels are assembled as homo- and heterotetramers with Kir5.1, we investigated the expression of both proteins throughout development (*Hibino et al., 2004*; *Ishii et al., 2003*). While in control littermates the number of ON OL-Kir4.1$^+$ channels increased with age (*Figure 1A–C*), we observed significantly lower Kir4.1 and Kir5.1 protein levels in *Kcnj10* cKO mice expressing *cre* recombinase under control of the *Olig2* promoter (*Figure 1A–B*; note that persistent expression corresponds to intact AS Kir4.1 in cKO animals) (*Schüller et al., 2008*). To study specific Kir4.1 functions in OLs in vivo, we studied *Kcnj10* loss-of-function in OPCs and mature OLs. As shown (*Figure 1D*, *Figure 1—figure supplement 1A*), Kir4.1 staining was substantially reduced from Apc$^+$ OL cell bodies but not Gfap$^+$ astrocyte fibers in ON samples from adult *Olig2-cre:Kir4.1$^{fl/fl}$* (cKO-1) and *Cnp-cre: Kir4.1$^{fl/fl}$* (cKO-2) mice (*Figure 1D*) confirming robust knockout efficiency in OLs (*Djukic et al., 2007*; *Schüller et al., 2008*; *Lappe-Siefke et al., 2003*). Levels of *Kcnj10* transcripts were higher in

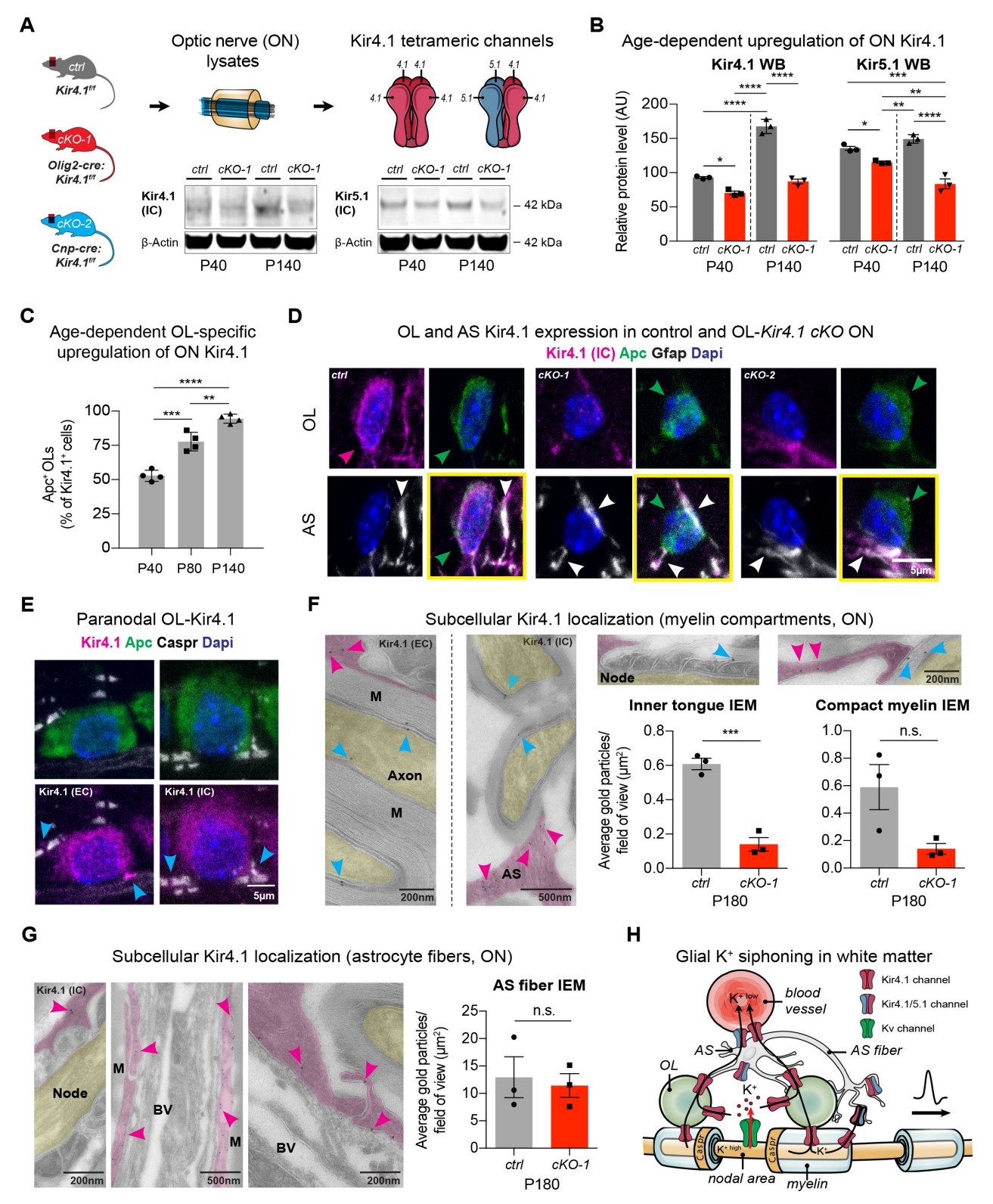

**Figure 1.** OL-Kir4.1 is upregulated during postnatal development and localized to peri-axonal spaces. Kir4.1 ON protein levels were upregulated between age P40 and P140, whereas Kir5.1 protein levels did not change during aging (A–B). Note substantial loss of Kir4.1 protein in *Olig2-cre* driven *Kcnj10* cKO (cKO-1) mice at P40, which became more apparent at P140; Kir5.1 protein was also reduced in cKO-1 ONs at P40 and P140 (control and cKO-1: n = 3 for all time points) (A–B). Quantification of Kir4.1[+] Apc[+] OLs confirmed age-dependent upregulation of OL-Kir4.1 channels between P40

*Figure 1 continued on next page*

*Figure 1 continued*

and P140 (n = 4 for all time points) (**C**). One-way ANOVA with Tukey's multiple comparison tests were performed in **B** and **C**; *p≤0.05, **p≤0.01, ***p≤0.001, ****p≤0.0001. Kir4.1 channels were lost from both ON OL cell bodies in cKO-1 and *Cnp-cre* driven *Kcnj10* cKO (cKO-2) mice versus controls (**D**). Note that Kir4.1+ OL are marked by magenta-colored arrowhead; Apc+ OLs are indicated by green arrowheads. Note AS Kir4.1 immunoreactivity and contacts of Kir4.1+ AS fibers with OLs (white arrowheads). Merged images are shown in panels highlighted by yellow surroundings (**D**). Kir4.1 was strongly expressed in OLs along spinal fiber tracts; note that cyan-colored arrowheads mark juxta-axonal Kir4.1 IR (**E**). Kir4.1 immunogold electron microscopy (IEM) labeling revealed presence of gold particles at inner and outer myelin tongue (cyan-colored arrowheads) and within AS fibers (magenda-colored arrowheads) adjacent to myelin sheaths (M = myelin) and blood vessels (BV = blood vessel; ctrl: n = 3, cKO-1: n = 3; **F–G**). Axon structures are highlighted in yellow, AS fibers are highlighted in magenta. Note decrease in inner tongue (**F**) but not compact myelin (**F**) or AS fiber (**G**) IEM labeling in cKO-1 ON tissue versus controls. Cartoon highlights proposed mechanism of glial K+ siphoning from axons during saltatory conduction towards blood vessels via a network of axonal Kv and glial Kir4.1 channels (**H**). Mann-Whitney tests were performed in **F–G**; ***p≤0.001, p=0.06 (**F**, compact myelin IEM), p=0.74 (**G**, AS fiber IEM). Data are presented as mean ±s.e.m in **B–C** and **F–G**.

DOI: https://doi.org/10.7554/eLife.36428.002

The following figure supplements are available for figure 1:

**Figure supplement 1.** Validation of OL-encoded *Kcnj10* cKO efficiency

DOI: https://doi.org/10.7554/eLife.36428.003

**Figure supplement 2.** Early developmental changes in OL-encoded *Kcnj10* loss-of-function

DOI: https://doi.org/10.7554/eLife.36428.004

myelinating OLs compared to OPCs in vitro, whereas *Kcnj16* mRNA decreased during OL maturation (***Figure 1—figure supplement 1B–C***) (***Kalsi et al., 2004***; ***Nwaobi et al., 2014***). Notably, we observed higher *Cacna1c* mRNA levels in the setting of loss-of-OPC-encoded *Kcnj10 in-vitro* (***Figure 1—figure supplement 1D***) (***Paez et al., 2010***; ***Cheli et al., 2015***). *Cacna1c* encodes Cav1.2, a major voltage-gated Ca²⁺ channel in OPCs. These findings demonstrate gradual upregulation of OL-Kir4.1 during postnatal development and suggest a more critical role of the channel later in life.

To understand spatial expression of OL-Kir4.1 channels in WM tracts, we studied longitudinal ON and spinal cord sections by high-resolution confocal microscopy and performed immunogold labeling of Kir4.1 channels in ON sections by electron microscopy (***Figure 1E–F***). Using antibodies either against an extracellular or intracellular Kir4.1 epitope, we could determine that OL-Kir4.1 channels are localized towards perinodal and juxta-axonal regions such as the inner tongue of myelin sheaths. Quantification of immunogold particles in ON sections using the Kir4.1 antibody against the intracellular epitope of the channel revealed a substantial decrease in inner but not outer tongue labeling in both *Olig2-cre* and *Cnp-cre* cKO lines as compared to controls (***Figure 1F***, ***Figure 1—figure supplement 1E–F***). Using a no-antibody control labeling, we could confirm specificity of the Kir4.1 labeling to myelin compartments and AS fibers (***Figure 1—figure supplement 1G***).

In WM AS, Kir4.1 immunogold particles were abundant in fibers and particular seen in processes in contact to the outer myelin tongue, in perivascular end feet and adjacent to the node of Ranvier (***Figure 1F–G***). Thus, the age-dependent and specific spatial expression pattern of glial Kir4.1 channels provides more evidence for a role of those channels in extracellular K+ buffering during electric activity along WM fiber tracts (***Figure 1H***).

## OL-Kir4.1 regulates but is not required for OL differentiation and early postnatal myelination

To investigate a role of OL-Kir4.1 in OL development, we compared OPC differentiation under control and *Kcnj10* loss-of-function conditions. We found that OPCs lacking *Kcnj10* exhibited precocious cell cycle exits as shown by decreased numbers of dividing cells (***Figure 1—figure supplement 2A–B***) and earlier onset of myelin production (***Figure 1—figure supplement 2C–D***). In addition, we found decreased mRNA levels of cell cycle and progenitor cell markers, such as *Uhrf1* and *Nkx2-2* in OPCs and conversely observed increased myelin basic protein (*Mbp*) mRNA levels in OLs in vitro (***Figure 1—figure supplement 2E–F***) (***Magri et al., 2014***). Mice deficient in OL-encoded *Kcnj10* had normal g-ratios in ON tissue at P40 (***Figure 2A–B***); however, they showed slightly smaller g-ratios in spinal WM tracts corresponding to thicker myelin sheaths (***Figure 2—figure supplement 1B–C***). Axonal diameters were not different in both ON and spinal WM samples at P40 (***Figure 2C***, ***Figure 2—figure supplement 1C***). The morphology of intra-axonal mitochondria with respect to circularity and density of mitochondria was not different in the ON and spinal cord WM between control

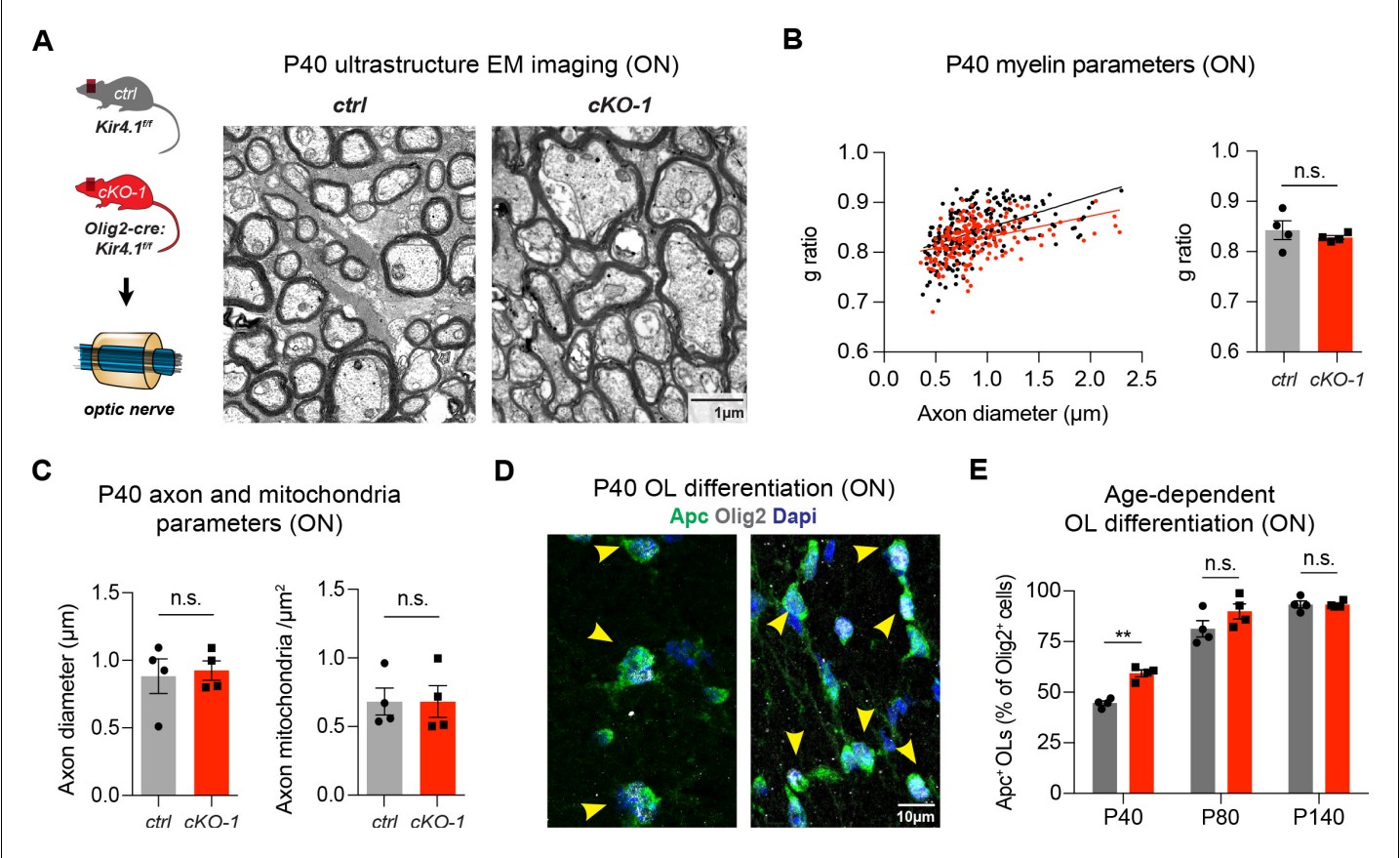

**Figure 2.** OL-Kir4.1 regulates early OL differentiation but is dispensable for myelination. Early developmental loss of OL-*Kcnj10* did not affect myelin sheath thickness or axon diameters in ONs from animals at P40 (210 axons from 4 control mice, 202 axons from 4 cKO-1 mice; **A–C**). Densities of intra-axonal mitochondria were not different between control and cKO-1 ONs at P40 (81 axons from 4 control mice, 77 axons from 4 *Kcnj10* cKO-1 mice; **C**). Mann-Whitney test was performed in **B–C**; p=0.49 (g-ratios, **B**), p=0.89 (axon diameter, **C**) and p=0.89 (mitochondria density, **C**). Immunostaining for Olig2 (pan-lineage marker for OPC/OL cells) and Apc (OL maturation marker) demonstrated precocious OL differentiation in cKO-1 ONs at P40 versus P80 and P140 (**D–E**). Two-way ANOVA with Sidak's multiple comparison test was performed in **E**; **p≤0.01. Data are presented as mean ±s.e.m in **B**, **C** and **E**.

DOI: https://doi.org/10.7554/eLife.36428.005

The following figure supplement is available for figure 2:

**Figure supplement 1.** Early white matter changes in OL-encoded *Kcnj10* loss-of-function

DOI: https://doi.org/10.7554/eLife.36428.006

and *Kcnj10 cKO* mice at P40 (*Figure 2C*, *Figure 2—figure supplement 1A,D*). Notably, the density of mature Apc⁺ Olig2⁺ OLs was transiently higher in *Kcnj10* cKO mice at P40, however, normalized during adulthood (*Figure 2D–E*). These results suggest that OL-Kir4.1 is involved in regulating cell cycle exit and OL differentiation during early postnatal development but is not a requirement for normal myelination.

## OL-*Kir4.1* is critical for normal motor and visual function in the adult CNS

We next investigated roles of OL-Kir4.1 during adulthood for maintenance of WM integrity in *Olig2-cre* driven *Kcnj10*-cKO-1 and *Cnp-cre* driven *Kcnj10*-cKO-2 mice and corresponding littermate controls up to 6 months of age. Both *Kcnj10*-cKO lines exhibited progressive neurological symptoms including abnormal gait and ataxia (*Video 1*), generalized seizures (*Video 2*) and hindlimb clasping (*Videos 3* and *4*) starting as early as three months of age. Early lethality by 6 months of age, most likely due to complications of epileptic seizures, was observed in ~70% of *Cnp-cre* and ~40% in *Olig2-cre* driven *Kcnj10* cKO mice (*Figure 3A*) (*Larson et al., 2018*). Higher seizure frequencies were

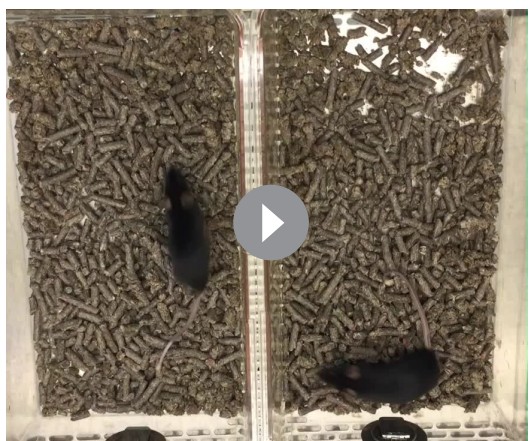

**Video 1.** Ataxia and motor dysfunction are progressive symptoms in OL-*Kcnj10* cKO mice. Video shows gait ataxia in OL-*Kcnj10* cKO mouse (left) as compared to littermate control (right) at P140.
DOI: https://doi.org/10.7554/eLife.36428.008

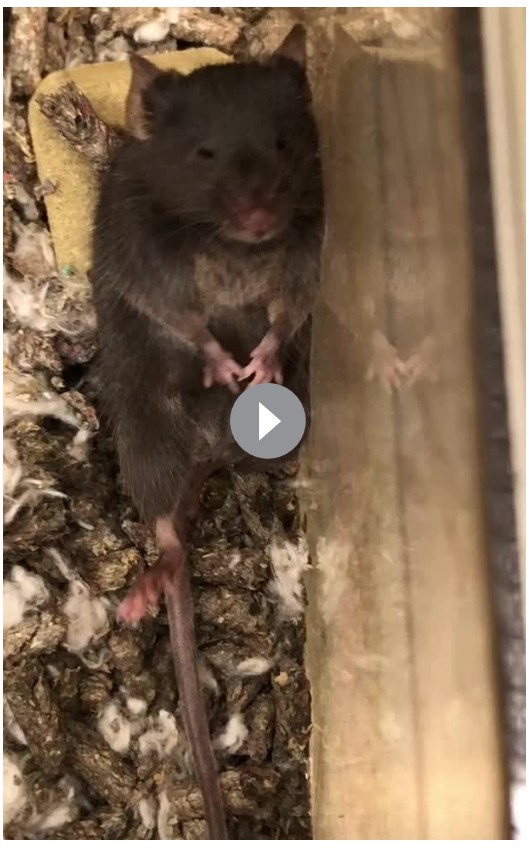

**Video 2.** Seizures are common and progressive in adult OL-*Kcnj10* cKO mice. Video shows generalized seizure in OL-*Kcnj10* cKO mouse at P140.
DOI: https://doi.org/10.7554/eLife.36428.009

a common feature of both cKO lines with increasing age, in keeping with prior findings (*Djukic et al., 2007*; *Larson et al., 2018*). All cKO mice showed reduced body weights versus control littermates at P140 (*Figure 3B*), and we found motor dysfunction as shown by reduced rotarod performance in both cKO cohorts at P140 (*Figure 3C*).

To investigate visual functions in live animals, we recorded single-flash light visual evoked potentials (VEPs) from anesthetized mice to measure conduction along the visual system. VEP has proven to be a useful method to study ON function in neurological patients with optic neuritis, such as multiple sclerosis (MS) (*Ridder and Nusinowitz, 2006*; *Graham and Klistorner, 2017*; *Kraft, 2013*). Here, both *Kcnj10* cKO lines had delayed VEP latencies at P140 (*Figure 3D–E*) indicating dysfunction in the visual system. In addition, we studied structural changes in the retina of live animals utilizing optical coherence tomography (OCT), where the thickness of specific retinal layers can be used as a surrogate for retinal ganglion cell (RGC) survival. Indeed, OCT imaging showed thinning of inner retinal layers (IRL) in cKO mice at P140 revealing a reduction in the number of ganglion cells and their corresponding axons (*Figure 3F*) (*Cruz-Herranz et al., 2016*). In summary, permanent loss-of-*Kcnj10* function in OLs results in visual dysfunction with retinal atrophy in adult mice as well as motor deficits and early mortality.

## OL-Kir4.1 is required for maintenance of WM integrity and late neuronal survival

We observed decreased Mbp protein levels, a critical myelin component, in ON and spinal cord lysates from 6 month old *Kcnj10* cKO mice, which could explain the delayed VEP latencies observed in older animals (*Figure 4A*, *Figure 4—figure supplement 1B–C*). A substantial loss of Kir4.1 protein could be detected in *Kcnj10* cKO ON and spinal cord tissue at P180 confirming robust knockout efficiency at that age (*Figure 4A*, *Figure 4—figure supplement 1A–C*). Additionally, at ultrastructural level, we observed disorganization of WM tracts (*Figure 4—figure supplement 1D*) and loss of myelin compactness in a subset of axons as well as evidence for axonal degeneration in *Olig2-cre* driven *Kcnj10* cKO ON and spinal cord tissue at P140 (*Figure 4B*, *Figure 4—figure supplement 1E*). Are

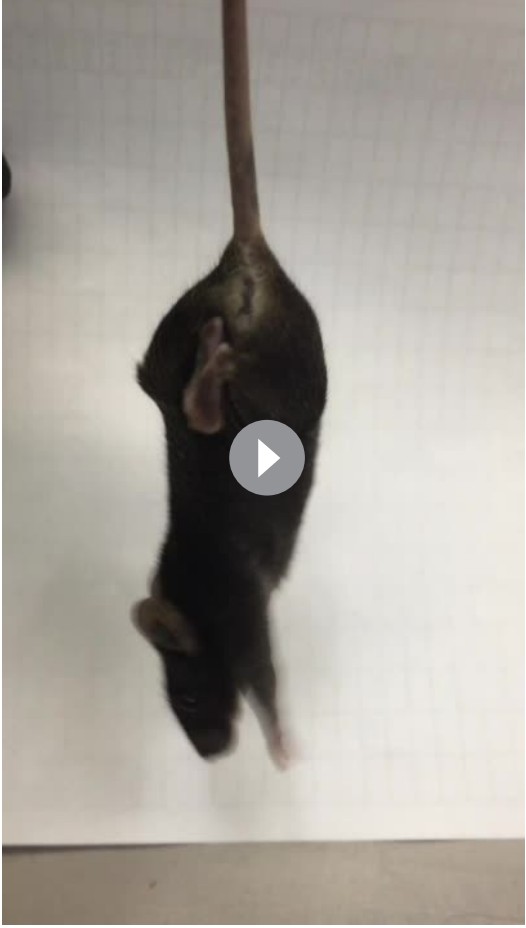 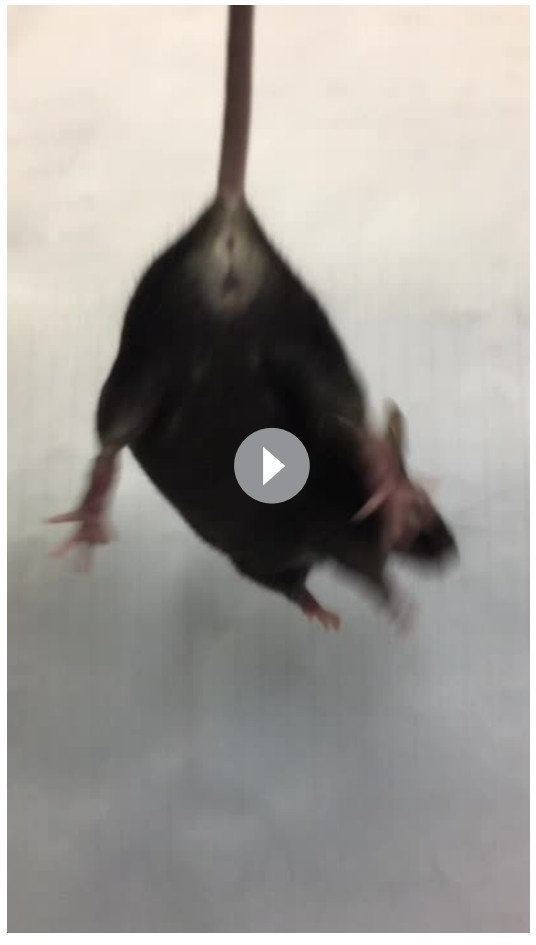

**Video 3.** Hind limb clasping is characteristic in adult OL-*Kcnj10* cKO mice. *Video 3* shows hind limb clasping as typical sign of motor dysfunction in OL-*Kcnj10* cKO mice compared to *Video 4* without presence of hind limb clasping in a control mouse at P140.
DOI: https://doi.org/10.7554/eLife.36428.010

**Video 4.** Hind limb clasping is not typical in normal adult mice. *Video 4* shows that hind limb clasping is not typical in an adult control mice.
DOI: https://doi.org/10.7554/eLife.36428.011

these results OL-specific or general to glial encoded-*Kir4.1* function? To address this, we investigated *Aldh1l1-cre* driven *Kcnj10* cKO that specifically deleted the *Kir4.1$^{fl/fl}$* allele in AS of the spinal cord (*Kelley et al., 2018*). In contrast to findings above, those AS-specific cKO mice did not show spinal WM tract abnormalities (*Figure 4—figure supplement 1E*) indicating specificity of OL-mediated Kir4.1 functions.

We further characterized WM pathologies and examined morphological features of ON and spinal WM intra-axonal mitochondria by electron microscopy. At P140, we noted enlarged axons in *Olig2-cre* driven *Kcnj10* cKO ONs (*Figure 4B,C*) with g-ratios not different between control and cKO mice (*Figure 4—figure supplement 1F*). Densities and total counts of intra-axonal mitochondria did not change in cKO ONs as compared to controls (*Figure 4—figure supplement 1G*). However, we found evidence for mitochondrial swelling as shown by increased mitochondrial circularity in *Olig2-cre* driven *Kcnj10* cKO ONs (*Figure 4C*). Immunostaining for neurofilaments (NFs) demonstrated loss of normally phosphorylated NFs (P- NF-H, SMI312[+]) in *cKO* ONs (*Figure 4D*) and an increase in non-phosphorylated NFs (non-P-NF-H, SMI32[+]) (*Figure 4E*), indicative of axon damage and increase in dystrophic axon numbers (*Trapp et al., 1998*; *Schirmer et al., 2011*). Axon pathology was accompanied by Iba1[+] microglia activation in ONs from *Olig2-cre* driven *Kcnj10* cKO animals (*Figure 4F–G*) and increased Gfap[+] staining (*Figure 4F,H*). We confirmed loss of Brn3a[+] RGCs through Brn3a staining of retinal whole mount preparations from both *Kcnj10* cKO lines at P180 (*Figure 4I–J*), consistent

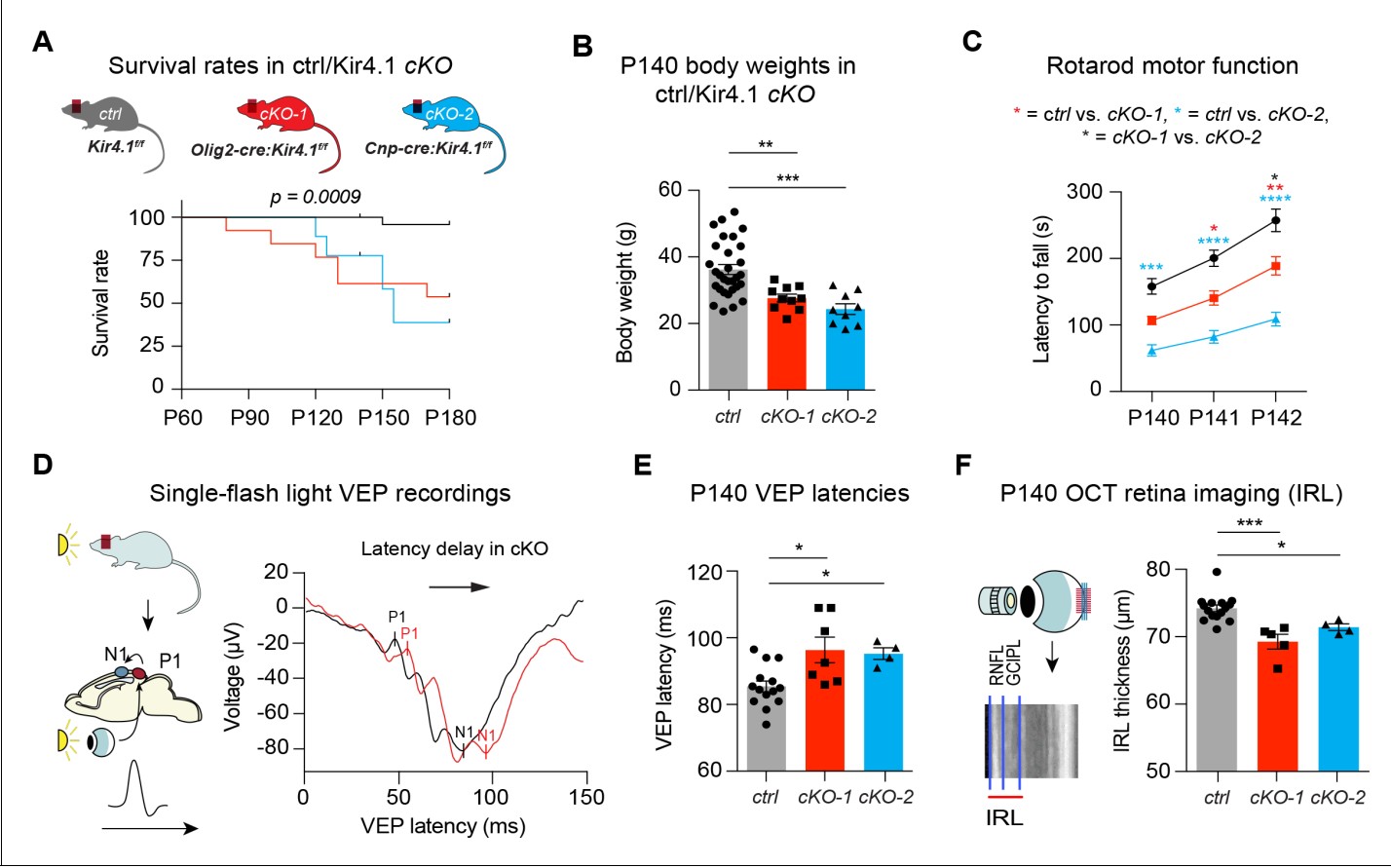

**Figure 3.** OL-Kir4.1 controls motor performance and visual function in adult mice. Mice lacking OL-Kir4.1 channels had increased mortality with survival of 96% in the control group (n = 29), 54% in cKO-1 (n = 13) and only 33% in cKO-2 (n = 9) mice at P180 (**A**). Log-rank (Mantel-Cox) test was performed and p-value shown in **A**. *Kcnj10* cKO-1 (n = 10) and cKO-2 (n = 9) mice were significantly smaller than control littermates (n = 30) at P140 (**B**). Kruskal-Wallis with Dunn's multiple comparisons test was performed in **B**; **p≤0.01, ***p≤0.001. Motor dysfunction with reduced rotarod performance has been observed in both cKO-1 (n = 7) and cKO-2 (n = 7) mice as compared to controls (n = 21) (**C**). Two-way ANOVA with Tukey's multiple comparisons test was performed in **C**; *p≤0.05, **p≤0.01, ***p≤0.001, ****p≤0.0001. Visual function was measured by single-flash light VEP recordings from control and *Kcnj10* cKO mice (**D–E**). VEPs were delayed in cKO-1 (n = 7) and cKO-2 (n = 4) mice versus controls (n = 14) (**E**). Kruskal-Wallis with Dunn's multiple comparisons test was performed in **E**; *p≤0.05. Retina integrity was measured by OCT imaging at P140 and revealed IRL thinning in cKO-1 (n = 5) and cKO-2 (n = 4) mice as compared to controls (n = 15; **F**). Kruskal-Wallis with Dunn's multiple comparisons test was performed in **F**; *p≤0.05, ***p≤0.001.

DOI: https://doi.org/10.7554/eLife.36428.007

with the OCT data (see above) indicating retrograde degeneration along the ON (*Figure 4K*) (*Raff et al., 2002*). In addition, by immunoreactivity retinal Gfap, Aqp4 and Iba1 levels were not significantly altered (*Figure 4—figure supplement 2*). However, we found increased Kir4.1 immunoreactivity in AS and Müller glia in *Olig2-cre* driven *Kcnj10* cKO retinae, which could reflect reactive upregulation of Kir4.1 channels in those glial cells (*Figure 4—figure supplement 2*). Together, these findings demonstrate that OL-Kir4.1 plays a major role in long-term neuro-axonal maintenance and integrity of long WM tracts of the ON and spinal cord.

## OL-*Kir4.1* is essential for WM integrity after chronic but not acute demyelinating injury

To study the role of OL-Kir4.1 during acute and chronic remyelination after focal white matter injury, we utilized the lysolecithin glial toxic injury model in the spinal cord at P80 (*Fancy et al., 2009*; *Fancy et al., 2011*; *Franklin and Ffrench-Constant, 2017*). In the acute situation, 14 days post-lysolecithin-induced focal demyelination of spinal WM tracts (14 dpl, corresponding to P94), we found that axons in lesions from *Olig2-cre* driven *Kcnj10* cKO animals harbored more mitochondria, which

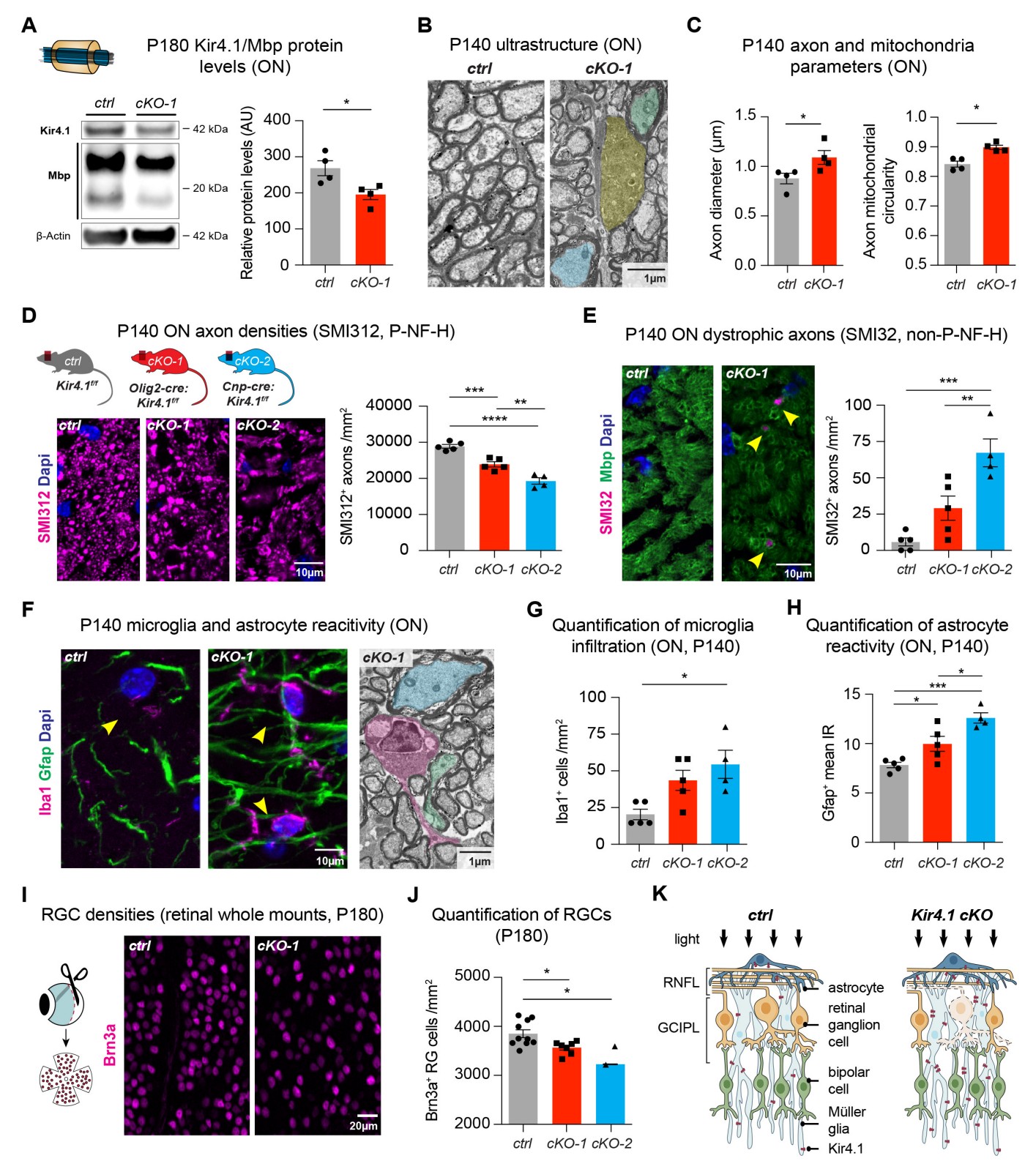

**Figure 4.** OL-Kir4.1 has a critical role in WM integrity and long-term maintenance. Myelin basic protein (Mbp) was decreased in ON lysates from cKO-1 (n = 4) mice versus controls (n = 4) at P180 (**A**). Mann-Whitney test was performed in **A**; *p≤0.05. Transmission electron microscopy demonstrated WM pathology with presence of degenerating (highlighted in yellow) and damaged axons of mild (highlighted in blue) and more pronounced severity (green highlight) at P140 (**B**). Axons were larger in cKO-1 versus control ONs at P140 (146 axons from 4 control mice, 139 axons from 4 cKO mice; **C**).
*Figure 4 continued on next page*

*Figure 4 continued*

Intra-axonal mitochondria were more circular as a proxy for swelling and dysfunction in cKO-1 mice ONs as compared to controls at P140 (86 axons from four control mice, 73 axons from 4 cKO mice; **C**). Mann-Whitney tests were performed in **C**; *p≤0.05. Numbers of physiological SMI312[+] (phosphorylated neurofilaments) axon profiles were reduced in ONs from cKO-1 (n = 5) and cKO-2 (n = 4) versus controls (n = 5; **D**), and numbers of dystrophic SMI32[+] (non-phosphorylated neurofilaments) axons were increased in ONs from cKO-2 (n = 4) versus control (n = 5) and cKO-1 (n = 5) mice (**E**). Kruskal-Wallis with Dunn's multiple comparisons tests were performed in **D** and **E**; **p≤0.01, ***p≤0.001, ****p≤0.0001. Iba1[+] microglia activation was a common feature in ONs from cKO-1 (n = 5) and cKO-2 (n = 4) versus control (n = 5) mice (**F–G**). Kruskal-Wallis with Dunn's multiple comparisons test was performed in **G**; *p≤0.05. Note microglial cell (highlighted in magenta) adjacent to dystrophic axons in cKO-1 ON (**F**). Astrogliosis as indicated by increased Gfap IR was enhanced in cKO-1 (n = 5) and cKO-2 (n = 4) ONs versus controls (n = 5) (**H**). One-way ANOVA with Tukey's multiple comparison test was performed in **H**; *p≤0.05, ***p≤0.001. Reduced densities of Brn3a[+] RGCs in cKO-1 (n = 7) and cKO-2 (n = 2) versus control mice (n = 10) were indicative of retrograde retinal neurodegeneration (**I–J**). Kruskal-Wallis with Dunn's multiple comparisons test was performed in **J**; *p≤0.05. Cartoon highlights pathological changes in the retina of *Kcnj10* cKO versus control mice with retrograde 'dying back' degeneration of RGCs and compensatory upregulation of Kir4.1 in retinal glial cells of *Kcnj10* cKO mice (**K**). Data are presented as mean ±s.e.m in **A**, **C**, **D–E**, **G–H** and **J**.

DOI: https://doi.org/10.7554/eLife.36428.012

The following figure supplements are available for figure 4:

**Figure supplement 1.** Long-term white matter pathologies in chronic OL-encoded *Kcnj10* loss of function

DOI: https://doi.org/10.7554/eLife.36428.013

**Figure supplement 2.** Long-term retinal changes in chronic OL-encoded *Kcnj10* loss-of-function

DOI: https://doi.org/10.7554/eLife.36428.014

were significantly more swollen/circular than their control counterparts (*Figure 5A–B*). However, we observed that early remyelination efficiency and axon diameters were not altered with similar g-ratios in WM lesions from *Olig2-cre* driven *Kcnj10* cKO mice and control littermates 14 dpl (*Figure 5C*, *Figure 5—figure supplement 1A*).

In contrast, in chronic lesions 60 days post-lysolecithin lesioning (60 dpl, corresponding to P140) mitochondrial and axon pathologies were readily detectable in *Olig2-cre* driven *Kcnj10* cKO mice with increased numbers of swollen intra-axonal mitochondria (*Figure 5D–E*) and enlarged axons (*Figure 5F*), whereas remyelination as measured by g-ratios was not affected (*Figure 5—figure supplement 1B*). These results indicate that OL-Kir4.1 is dispensable for (re)myelination but that its function is crucial for axon support and maintenance after long-term demyelinating WM injury.

## Discussion

Although support of axons by myelinating oligodendrocytes is an essential requirement for long-term maintenance of function in the CNS, precise mechanisms of OL-axon trophic interactions are incompletely understood. Here, we studied the role of OL-Kir4.1 channels for WM integrity and maintenance in the ON and spinal WM tracts during postnatal development, adulthood and WM injury. Both fiber tracts are composed of long axons that rely on a strong glial support establishing proper action potential propagation (*Nave, 2010*). Long fiber tracts are particularly vulnerable to WM pathologies as observed in MS and EAST/SeSAME syndrome, and the anterior visual system comprising the retina and the ON is a common lesion site in MS. Also, the system is easily accessible to precise imaging and measurements such as OCT and VEP to monitor neuro-axonal function (*Green et al., 2017*; *Ontaneda et al., 2017*).

Interestingly, we found that Kir4.1 channels were localized to perinodal OLs and within myelin in juxta-axonal spaces along the ON, suggesting roles of Kir4.1 channels for proper axonal function (*Hibino et al., 2004*; *Bay and Butt, 2012*; *Ishii et al., 2003*; *Kalsi et al., 2004*). To our knowledge, this is the first report of a mature OL-associated K[+] channel with polar expression oriented towards axons, i.e. localization at the inner myelin tongue. Because the lactate transporter MCT1 also shows a similar juxta-axonal expression pattern (*Lee et al., 2012*), it is possible that other ion channels and solute carrier transporters are arrayed in a similar way to maintain axon energy, activity and integrity. Thus, we propose that Kir4.1 comprises a 'myelin nanochannel'. While this positioning is consistent with a role in siphoning K[+] within myelin segments, further studies are needed to confirm this function.

Our loss-of-function studies identified two temporally-regulated roles for OL-Kir4.1. Using both early-acting *Olig2-cre* or *CNP-cre*, which initiates expression at a later stage, we defined early and later functions of Kir4.1 and ruled out caveats associated with heterozygous effects of *cre* knockin to

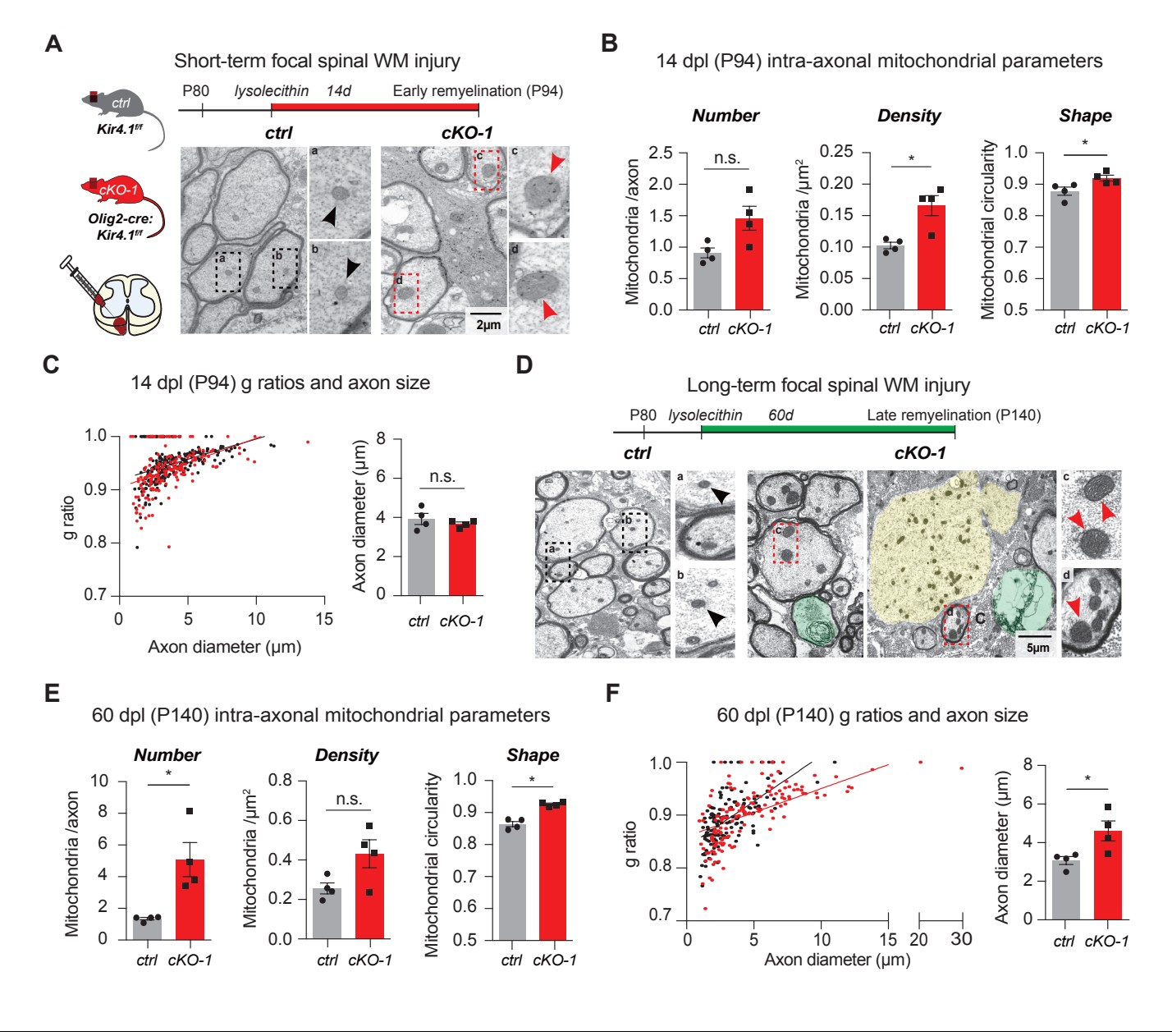

**Figure 5.** OL-Kir4.1 is dispensable for remyelination, but critical for long-term axon maintenance after WM demyelinating injury. OL-Kir4.1 function was studied in short- (A–C) and long-term remyelination (D–F) after lysolecithin-induced focal demyelination to ventrolateral spinal WM tracts. Mice were euthanized and perfused at two survival time points corresponding to days post lesioning (dpl, n = 4 for each time point and genotype): 14 dpl (corresponding to P94, representing new myelin sheath formation) and 60 dpl (corresponding to P140, full remyelination). Densities of intra-axonal mitochondria were increased in cKO-1 (176 axons from 4 mice) versus control animals (230 axons from 4 mice) at 14 dpl and circularity/swelling of intra-axonal mitochondria was higher in cKO-1 (79 axons from 4 mice) versus controls (141 axons from four mice; A-B). Note high-magnification images of representative mitochondria in A indicating enlarged mitochondria in axons from cKO-1 versus control lesioned tissue. Conversely, loss of OL-Kir4.1 did not affect g-ratios and axon diameters in cKO-1 (176 axons from 4 mice) versus control mice (230 axons from four mice) (C). At 60 dpl, cKO-1 mice exhibited pronounced WM damage during long-term remyelination with presence of enlarged and degenerating axons (highlighted in green and yellow) as well as increased numbers of swollen intra-axonal mitochondria (D). Numbers of intra-axonal mitochondria were increased in cKO-1 versus control mice 60 dpl but densities of mitochondria were not different due to enlargement of lesion axons and thus relative lower mitochondria densities in cKO-1 axons; intra-axonal mitochondria were more circular in cKO-1 (90 axons from 4 mice) mice as compared to controls (88 axons from 4 mice) at 60 dpl (E). Remyelination was efficient and not different between cKO-1 (139 axons from 4 mice) and control mice (152 axons from 4 mice) at 60 dpl, however, enlarged axons were observed in cKO-1 mice as compared to controls (F). Mann-Whitney tests were performed in B–C and E–F; *p≤0.05, p=0.69 (C). Data are presented as mean ±s.e.m in B–C and E–F.

*Figure 5 continued on next page*

*Figure 5 continued*

DOI: https://doi.org/10.7554/eLife.36428.015

The following figure supplement is available for figure 5:

**Figure supplement 1.** Short- and long term remyelination efficiencies in chronic OL-*Kcnj10* loss-of-function

DOI: https://doi.org/10.7554/eLife.36428.016

the *Olig2* and *Cnp* loci . Regarding early developmental requirements of OL-Kir4.1 function, the cKO resulted in somewhat precocious maturation and myelination. This suggests either a direct role in regulating differentiation or indirect effects of altered ion currents on voltage-gated $Ca^{2+}$ channels in *Kcnj10*-deficient OPCs (*Paez et al., 2010*; *Cheli et al., 2015*; *Cheli et al., 2016*). Based on observations during development and regeneration, we conclude that Kir4.1 regulates, but is not required for normal myelination. Other studies have indicated early roles for OL-Kir4.1 channels in support of acute axonal activity (*Larson et al., 2018*).

Secondly, we studied late (adult) functions of glial Kir4.1 and identified a specific OL-related role of Kir4.1 channels for support of long axons of the ON and the spinal WM. Indeed, our data demonstrated that OL-mediated Kir4.1 function was essential in WM maintenance, a function not found in AS-specific *Kcnj10* cKO mice (*Kelley et al., 2018*). While in principle, loss of OL-Kir4.1 currents can be compensated for by OPC- or AS-encoded Kir4.1/Kir5.1 channels, or homotetrameric Kir5.1 channels with PSD-95 (*Hibino et al., 2010*; *Tanemoto et al., 2002*), it is evident that such compensation eventually fails during adulthood in OL-specific *Kcnj10* cKO mice, possibly due to increasing demand for perinodal $K^+$ buffering during sustained electric activity, such as with physical exertion (rotarod performance) or after (excitotoxic) demyelinating WM injury. Indeed, we found that Kir4.1 (*Kcnj10*) expression increases in OLs during maturation and aging, and consistent with progressive requirements in adults, it has been reported that Kir4.1-mediated $K^+$ buffering becomes more important during ON high-frequency stimulation and that OL-*Kcnj10* loss-of-function increases neuronal excitability in adult mice (*Bay and Butt, 2012*; *Larson et al., 2018*). In contrast, mice without AS-encoded *Kcnj10* lack spinal WM tract abnormalities typical of OL-encoded *Kir4.1* cKO mice. Thus, our data parses out the specific roles of OL-Kir4.1 and reveals a new supportive role of OLs for long-term axonal maintenance.

Overall, our findings suggest a model in which developmental upregulation of OL-Kir4.1 is linked to increasing needs for axonal $K^+$ homeostasis during adulthood. Siphoning of $K^+$ is most likely established through an orchestrated action of juxta-axonal and perinodal glial Kir4.1 channels. The ad-axonal expression pattern of mature OL-Kir4.1 channels emphasizes that those channels might have an essential role in $K^+$ clearance from axons, a function that becomes more critical later in life. However, we cannot rule out other distinct functions of Kir4.1 that are more linked to cell body-associated localization of Kir4.1 and OL-intrinsic functions like differentiation during early postnatal development (see above). Indeed, motor dysfunction and spastic paraplegia are progressive age-related neurological symptoms in individuals with EAST/SeSAME syndrome (*Cross et al., 2013*), which would resemble features that come with permanent knockout of mature OL-Kir4.1 channels resulting in mitochondrial damage and progressive neurodegeneration. Increased numbers and swelling of mitochondria as well as enlarged axon diameters were early indicators of impending axonal degeneration in OL-*Kcnj10* cKO mice, especially when challenged with focal WM demyelination. Of note, morphological changes of intra-axonal mitochondria have been associated with lysolecithin-induced demyelinating lesions (*Zambonin et al., 2011*; *Kiryu-Seo et al., 2010*) and axonal degeneration in experimental autoimmune encephalomyelitis (EAE), an animal model of MS (*Nikić et al., 2011*). In the latter, it might characterize a phenomenon that is reversible and precede ultimate axonal degeneration. Notably, higher seizure frequencies with increasing age might contribute to chronic cognitive and motivational dysfunction in cKO animals; however, it is unlikely that this has a major effect on degenerative changes in long white matter tracts.

In summary, we identified that OL-Kir4.1 channels are localized to myelin inner tongue and juxta-nodal regions suggesting functions in removal of $K^+$ commensurate with axonal activity. Moreover, we describe a novel role for OL-encoded *Kcnj10* in long-term maintenance of axons in WM tracts of the CNS. Permanent loss of OL-Kir4.1 channels resulted in progressive damage to axons and ultimately loss of neurons with increasing age. Our findings raise the question of whether similar

mechanisms might exist in Schwann cells that serve the myelinating function in the peripheral nervous system. Additionally, our finding might have relevance for understanding mechanisms of axonal degeneration in chronic MS lesions, where KIR4.1 channels are dysregulated and lost during lesion progression (*Schirmer et al., 2014*). If so, dissecting mechanisms of dysregulated $K^+$ homeostasis in chronic neuro-inflammatory conditions could help develop neuroprotective strategies designed to correct local peri-axonal $K^+$ imbalances.

## Materials and methods

### Mice

All mouse strains were maintained at the University of California, San Francisco (UCSF) specific pathogen-free animal facility under protocol number AN110094. All animal protocols were approved by and in accordance with the guidelines established by the Institutional Animal Care and Use Committee and Laboratory Animal Resource Center. *Kir4.1^{fl/fl}* mice were obtained from Ken D. Mc Carthy (University of North Carolina, Chapel Hill, NC, USA) and generated as previously described (*Kelley et al., 2018*; *Djukic et al., 2007*). *Olig2-tva-cre* transgenic mice were generated as previously described (*Schüller et al., 2008*). *Cnp-cre* transgenic mice were commercially available (Jackson Lab) and had been previously generated (*Lappe-Siefke et al., 2003*). *Aldh1l1-cre* mice were generated by the GENSAT project as previously described (*Tien et al., 2012*). All mice were maintained on a 12 hr light/dark cycle with food and water available *ad libitum*. All mice were kept on a C57BL/6J background and *Kir4.1^{fl/fl}* littermate controls were used for all experiments.

### Behavioral analysis

All behavioral experiments were performed at the UCSF Neurobehavioral Core for Rehabilitation Research. Rotarod (Ugo Basile) testing was performed on a rotating rod that accelerated from 0 to 40 rotations per minute (rpm) during a 5 min period. The latency to fall (in seconds) was recorded for each mouse in order to assess for motor deficits and endurance. Animals were tested three times per day for three consecutive days.

### Optical coherence tomography

Retinal in-vivo imaging using optical coherence tomography (OCT) was carried out as previously described (*Sagan et al., 2017*; *Sagan et al., 2016*). Briefly, pupils were dilated with 1% tropicamide (Akorn), and mice were anesthesized using isoflurane (Isothesia, Henry Schein Animal Health). Guided by the infrared fundus image, vertical and horizontal OCT scans confirmed that the retina lays perpendicular to the laser. Then, 25 B-scans in high-resolution mode were taken and rasterized from 30 averaged A-Scans using the Spectralis Diagnostic Imaging system with the TruTrack eye-tracker to avoid motion artifacts (Heidelberg Engineering). Automated segmentation was done using the modular imaging software (Heidelberg Eye Explorer). Retinal segments were manually corrected corresponding to the inner limiting membrane (ILM) and inner plexiform layer (IPL), representing the limits of the inner retinal layers (IRL). IRL thicknesses were calculated using the Early Treatment Diabetic Retinopathy Study (ETDRS) grid with diameters of 1, 2, and 3 mm centered on the optic disc, and exported into a spreadsheet file. Both eyes for each mouse were examined, using generalized estimating equations with an exchangeable correlation matrix and adjustments for intra-subject inter-eye correlations (*Cruz-Herranz et al., 2016*). All experiments were carried out by an operator blinded for mouse genotype and treatment condition.

### Visual evoked potentials

Visual pathway conduction was examined by recording of flash-light visual evoked potentials (VEP) using an Espion Diagnosys setup (Diagnosys). Mice were anesthetized using xylazine (20 mg/ml, Anased, Akorn) and ketamine (100 mg/ml, Ketathesia, Henry Schein Animal Health) in sterile 0.1M PBS through intraperitoneal (i.p.) injection. Mice were adapted to darkness for 5 min before placed in the recording system. The measuring electrode was a needle electrode placed medially in the area corresponding to the visual cortex, the reference electrode was placed under the nasal skin, while the grounding electrode was positioned at the tail root. VEP recordings started 13 min after i.p. injection and consisted of three runs (3 cd·s/m2, 1 Hz, 4 ms, 6500K, 100 sweeps). Mice VEP

results in a negative wave, corresponding to P100 in humans, after approximately 75 ms, called N1. After three recordings per mouse were collected, the latency was calculated as the average of the second and third N1 result. All experiments were carried out by an operator blinded for mouse genotype and treatment condition.

## Oligodendrocyte progenitor cell cultures

Isolation and purification of mouse OPCs was performed according to previously described immuno-panning protocols using an anti-Pdgfra (CD140a) antibody for positive selection of OPCs (*Shiow et al., 2017*; *Yuen et al., 2014*; *Dugas and Emery, 2013*). Briefly, OPCs were immuno-panned from P7-P9 mouse cortices and plated on poly-D-lysine coverslips (Neuvitro). Cells were kept in proliferation media (PDGF-AA, CNTF, and NT3; Peprotech) at 10% $CO_2$ and 37°C. After two days in proliferation media, differentiation was induced by changing media to contain CNTF and tri-iodothyronine (T3; Sigma). Note that mycoplasma contamination testing was negative.

## Focal white matter demyelinating injury

Mice were anesthetized using xylazine (20 mg/ml, Anased) and ketamine (100 mg/ml, Ketathesia) in sterile 0.1M PBS through i.p. injection. Focal demyelinating WM lesions were induced in the lower thoracic spinal cord around T12/13 according to previously published protocols (*Fancy et al., 2009*; *Fancy et al., 2011*). Briefly, 1 µL of 1% lysolecithin (l-a-lysophosphatidylcholine, Sigma) were injected to induce focal WM demyelination in the ventrolateral spinal cord of P80 *cKO* and control littermate mice.

## Mouse tissue immunohistochemistry

Mice were deeply anesthetized and transcardially perfused with ice-cold phosphate-buffered saline (PBS) followed by 4% paraformaldehyde (PFA) and subsequently post-fixed in PFA for 1 hr. After post-fixation, samples were cryoprotected in 30% sucrose in PBS for 48 hr at 4°C and embedded in optimal cutting temperature (OCT) compound (Tissue-Tek). 16 µm-cryosections were collected on superfrost slides (VWR) using a CM3050S cryostat (Leica) and fixed in either 4% PFA at room temperature (RT) or ice-cold methanol. Sections were blocked in 0.1M PBS/0.1% Triton X-100/10% goat/horse/donkey sera for 1 hr at RT. Primary antibody incubations were carried out overnight at 4°C. After washing in 0.1M PBS, cryosections were incubated with secondary antibodies diluted in 0.1M PSB/0.1% Triton X-100 for 2 hr, RT. For immunofluorescence, Alexa fluochrome-tagged secondary IgG antibodies (1:500, Invitrogen) were used for primary antibody detection. Slides with fluorescent antibodies were mounted with DAPI Fluoromount-G (SouthernBiotech). Negative control sections without primary antibodies were processed in parallel.

## Transmission electron microscopy

Tissue processing and image acquisition by transmission electron microscopy (EM) was carried out as previously reported (*Harrington et al., 2010*). Briefly, mice were perfused transcardially with 0.1M PSB followed by 4% glutaraldehyde and 0.008% $CaCl_2$ in 0.1M PBS. After post-fixation in glutaraldehyde, ON and spinal cord tissue blocks were further fixed in osmium tetroxide at 4°C overnight, dehydrated through ascending ethanol washes, and embedded in TAAB resin (TAAB Laboratories). 1 µm-thick sections were cut, stained with toluidine blue, and examined by light microscopy to assess WM integrity and identify lesions. Non-lesion ON, spinal cord and remyelinating lesion blocks were examined by transmission EM (Hitachi, H600), and g-ratio calculations of axons in the area of interest were calculated by dividing the diameter of an axon by the diameter of axon and associated myelin sheath. Between 100–200 axons per group of 4 animals were analyzed. Briefly, images of transverse ON and spinal cord sections were taken at either 6000x or 10,000x magnification. Digitized and calibrated images were analyzed, and linear regression was used to indicate the differences between *cKO* and control groups in myelin thickness across the range of axon diameters. Numbers and densities of intra-axonal mitochondria per axon area were quantified, and circularities of individual mitochondria were calculated using Fiji ImageJ software (NIH). Circularity is a two-dimensional sphericity index with a value of 1 corresponding to a perfect sphere: Circularity = $4\pi \times$ Area/(Perimeter)$^2$.

## Immunoelectron microscopy

Briefly, mice at the age of 6 months were perfused with 4% formaldehyde and 0.2% glutaraldehyde in 0.1 M phosphate buffer containing 0.5% NaCl. ONs were dissected and postfixed in the same fixation solution for 24 hr. Small pieces of ON tissue were embedded in 10% gelatine and subsequently infiltrated with 2.3 M sucrose in 0.1 M phosphate buffer overnight. Small blocks of gelatine containing the ON pieces were mounted on aluminum pins for ultramicrotomy and frozen in liquid nitrogen. Ultrathin cryosections were prepared with a 35° cryo-immuno diamond knife (Diatome) using a UC7 cryo-ultramicrotome (Leica). For immuno-labeling sections were incubated with antibodies directed against either intracellular or extracellular epitopes of Kir4.1, which were detected with protein A-gold (10 nm) obtained from the Cell Microscopy Center, Department of Cell Biology, University Medical Center Utrecht, The Netherlands. Sections were analyzed with a LEO EM912AB (Zeiss), and digital micrographs were obtained with an on-axis 2048 × 2048 CCD camera (TRS) (*Werner et al., 2007*). For quantification of immunogold labeling, 11 images with a size of 8 μm x 8 μm per animal were obtained from one optic nerve section in a systematic random sampling regime. The total analyzed field of view covers 704 $\mu m^2$ per animal. Of each genotype, three animals were analyzed after labeling with the antibody against the intracellular epitope of Kir4.1 (APC-035, Alomone labs) including a reagent control by omitting the primary antibody. All gold particles were counted and assigned to structures identified by morphology such as astrocyte profiles, compact myelin, myelin inner and outer tongue.

## Whole-mount immunohistochemistry

Retinal whole-mount preparations were performed from eyes that were post-fixed in 4% PFA for additional 24 hr after intracardial perfusion. Afterwards, eyes were kept in PBS, and retinal whole mounts were prepared as described previously (*Sagan et al., 2016*). Retinal ganglion cells were stained with an anti-Brn3a antibody and quantified using the Fiji ImageJ software.

## Immunocytochemistry

Proliferating OPCs were quantified by immunoreactivity for EdU or pH3. Stainings were performed on day three after immunopanning and one day after replacement of proliferating media. Mbp immunostaining was performed to quantify myelinating OLs after switching to differentiating culture conditions as previously described (*Shiow et al., 2017*).

## Proliferation assays

P1 pups were injected with 10 mg/ml Bromodeoxyuridine (BrdU) i.p. (BD Pharmingen). After two hours, mice were intracardially perfused. After tissue processing and cryo-sectioning, DNA on sections was denatured by incubation in 2N HCl for 30 min at 37°C, followed by rinses with 0.1M boric buffer. Then, BrdU incorporation was visualized performing IHC using an anti-BrdU antibody. For in-vitro proliferation studies, cells were incubated with 5-ethynyl-2′-deoxyuridine (EdU) for 1 hr, and EdU incorporation was visualized using the Click-iT EdU Kit (Invitrogen) according to the manufacturer's instructions.

## Quantitative polymerase chain reaction (qPCR)

For mouse OPC/OL mRNA analysis, RNA was extracted using Trizol (Invitrogen) and purified using the RNAeasy Kit (Qiagen) according to manufacturer's instructions. Complementary DNA (cDNA) was generated using the High-Capacity RNA-to-cDNA Kit (Applied Biosystems). qPCR was performed on a LightCycler 480 using LightCycler 480 SYBR Green I Master mix, and melting curves were analyzed to ensure primer specificity. Mouse primers used included *Kcnj10* (forward: AGAGGGCCGAGACGAT; reverse: TTGACCTGGTTGAGCCGAATA), *Kcnj16* (forward: CCTGTGTC TCCTCTTGAAGG; reverse: TGTGCTTAGGTGATACAATACGG), *Cacna1c* (forward: CCTAATGGG TTCGTTTCAGAAGT; reverse: TCCGGTTACCTCCAGGTCA), *Cdk1* (forward: GCCAGAGCG TTTGGAATACC; reverse: CAGATGTCAACCGGAGTGGAGTA), *Cdk2* (forward: GGCTCGACAC TGAGACTGAA; reverse: GGTGCAGAAATTCAAAAACCA), *Uhrf1* (forward: TGAAGCGGATGACAA-GACTG; reverse: CAGGGCTCGTCCTCAGATAG), *Nkx2-2* (forward: GCCTCCAATACTCCCTGCAC; reverse: GTCATTGTCCGGTGACTCGT), *Cnp* (forward: GGCGGCCCCGGAGACATAGTA; reverse: GCTTGGGCAGGAATGTGTGGC), *Mbp* (forward: CCCAAGGCACAGAGACACGGG; reverse: TACC

TTGCCAGAGCCCCGCTT) and *18* s (forward: GTTCCGACCATAAACGATGCC; reverse: TGGTGG TGCCCTTCCGTCAAT). For normalization, mRNA expression levels were calculated according to ribosomal *18* s expression and presented as relative mRNA levels throughout the figures.

## Western blot

Preparation of protein extracts, immunoblots and chemiluminescence detection was done as previously described (*Kenney and Rowitch, 2000*). Fluorescent detection of proteins was carried out using the Li-Cor Odyssey system (Li-Cor) according to the manufacturer's instructions. After blocking in PBS Odyssey Blocking Buffer (Li-Cor) for 1 hr at RT, primary antibodies were incubated overnight at 4°C. IRDye Goat anti-mouse and anti-rabbit (680 and 800) fluorescent secondary antibodies (Li-cor) were used for protein detection on the Odyssey Cxl imaging system.

## Antibodies

The following antibodies were used for immunopanning, immunocytochemistry, immunohistochemistry and Western Blot experiments: goat anti-OLIG2 (AF2418, R and D Systems, 1:50), mouse anti-APC (clone CC1, OP80, 1:300, Millipore Sigma), mouse anti-NOGO-A (clone 11C7, gift from M.E. Schwab, 1:3,000), rat anti-MBP (ab7349, Abcam, 1:500), mouse anti-MOG (clone 8–18 C5, 1:1,000, Millipore Sigma), rat anti-GFAP (clone 2.2B10, 13–0300, Invitrogen, 1:1,000), rabbit anti-AQP4 (AB3594, 1:500, Millipore Sigma), rabbit anti-KIR4.1 (APC035, Alomone Labs, 1:3,000), rabbit anti-KIR4.1 (APC-165, Alomone Labs, 1:1,000), rabbit anti-KIR5.1 (APC123, Alomone Labs, 1:500), mouse anti-Neurofilament H (NF-H), nonphosphorylated (clone SMI32, 801701, Biolegend, 1:10,000), mouse anti-Neurofilament H (NF-H), phosphorylated (clone SMI312, 837904, Biolegend, 1:1,000), rabbit anti-IBA1 (019–19741, Wako, 1:500), goat anti-BRN3a (sc-31984, Santa Cruz, 1:200), rabbit anti-KCNQ2 (ab22897, Abcam, 1:200) rabbit anti-CASPR (ab34151, Abcam, 1:1,000), mouse anti-BRDU (347580, BD Biosciences, 1:200), rabbit anti-phospho-Histone H3 (pH3, 9701, Cell Signaling, 1:500), rat anti-CD140a (558774, BD Biosciences, 1:500), mouse anti-β-ACTIN (A5316, Sigma, 1:7,000).

## Image acquisition and analysis

Bright field images were acquired on a Zeiss Axio Imager two microscope. Fluorescent images were taken using Leica TCS SP8 and TCS SPE laser confocal microscopes with either 10x, 20x, 40x or 63x objectives; all fluorescent pictures are Z-stack confocal images, unless stated otherwise. Images were processed using Fiji ImageJ or Photoshop software (Adobe) and exported to Illustrator vector-based software (Adobe) for figure generation.

## Statistical analysis

Data are presented as mean ±SE of mean (SEM). Analyses was performed using two-tailed parametric or non-parametric (Mann-Whitney, Kruskal-Wallis) t-tests for two groups if applicable, one-way ANOVA with corresponding post-hoc tests for multiple group comparisons and paired two-way ANOVA with post-hoc tests for longitudinal group comparisons at different time points. Kaplan-Meier estimator was used to quantify survival rates between transgenic mice during aging. Level of significance was determined as described in the individual figure legends. P values were designated as follows: *$p \leq 0.5$, **$p \leq 0.01$, ***$p \leq 0.001$, ****$p \leq 0.0001$. Analyses were performed using GraphPad Prism (GraphPad Software).

## Acknowledgments

We thank Ken D McCarthy (University of North Carolina, NC) for *Kir4.1-floxed* mice and Dwight E. Bergles (Johns Hopkins University, MD) as well as Detlef Bockenhauer (University College London, UK) for helpful comments and sharing unpublished results. We thank Jose L. Rodas-Rodriguez and Samir Elmojahid for excellent technical assistance. We thank Anna Hupalowska for assistance with figure illustrations. Behavioral data were obtained with the help of the UCSF Neurobehavioral Core for Rehabilitation Research. LS was supported by postdoctoral fellowships from the German Research Foundation (DFG, SCHI 1330/1–1) and the National Multiple Sclerosis Society Dave Tomlinson Research Fund (NMSS, FG-1607–25111). WM, BS and KAN were supported by the Cluster of

Excellence and the DFG Research Center for Nanoscale Microscopy and Molecular Physiology of the Brain (EXC171), a DFG collaborative research project (TR43) and the European Research Council (ERC advanced grant - AxoGLIA, ERC advanced grant - MyeliNANO). ACH was supported by a NMSS postdoctoral fellowship (FG-20102-A-1). CC was supported by a Training Fellowship from the Italian Multiple Sclerosis Society (FISM, Cod. 2013/B/4). AKP was supported by postdoctoral fellowships from the Swiss National Science Foundation (P2SKP3_164938/1, P300PB_177927).KWK was supported by the Medical Scientist Training Program and a California Institute of Regenerative Medicine pre-doctoral fellowship. The study was supported by the NMSS (DHR), the UK Multiple Sclerosis Society (RJMF, CZ), the Adelson Medical Research Foundation (DHR, RJMF, KAN), the Cambridge Biomedical Research Center (DHR), grants from the NINDS (NS040511) and the Wellcome Trust (to DHR).

## Additional information

### Competing interests

Klaus-Armin Nave: Reviewing editor, *eLife*. Lucas Schirmer: filed a patent for the detection of antibodies against KIR4.1 in a subpopulation of patients with multiple sclerosis (WO2015166057A1). The other authors declare that no competing interests exist.

### Funding

| Funder | Grant reference number | Author |
|---|---|---|
| National Multiple Sclerosis Society | FG-1607-25111 | Lucas Schirmer |
| Deutsche Forschungsgemeinschaft | SCHI 1330/1-1 | Lucas Schirmer |
| Schweizerischer Nationalfonds zur Förderung der Wissenschaftlichen Forschung | P2SKP3_164938/1 | Anne-Katrin Pröbstel |
| Associazione Italiana Sclerosi Multipla | 2013/B/4 | Christian Cordano |
| Multiple Sclerosis Society | Project grant | Robin JM Franklin |
| Wellcome Trust | Senior investigator award | David H Rowitch |
| Dr. Miriam and Sheldon G. Adelson Medical Research Foundation | Collaborative research grant | Klaus-Armin Nave Robin JM Franklin David H Rowitch |
| National Institutes of Health | NS040511 | David H Rowitch |
| California Institute of Regenerative Medicine | Medical Scientist Training Program | Kevin W Kelley |
| European Commission | ERC advanced grant - AxoGLIA | Klaus-Armin Nave |
| National Multiple Sclerosis Society | FG-20102-A-1 | Andrés Cruz-Herranz |
| Deutsche Forschungsgemeinschaft | EXC171 | Wiebke Möbius Klaus-Armin Nave |
| Schweizerischer Nationalfonds zur Förderung der Wissenschaftlichen Forschung | P300PB_177927 | Anne-Katrin Pröbstel |
| European Commission | ERC advanced grant - MyeliNANO | Klaus-Armin Nave |
| Deutsche Forschungsgemeinschaft | TR43 | Wiebke Möbius |

The funders had no role in study design, data collection and interpretation, or the decision to submit the work for publication.

## Author contributions

Lucas Schirmer, Conceptualization, Data curation, Software, Formal analysis, Funding acquisition, Validation, Investigation, Visualization, Methodology, Writing—original draft, Project administration, Writing—review and editing; Wiebke Möbius, Conceptualization, Formal analysis, Supervision, Funding acquisition, Validation, Investigation, Visualization, Methodology, Writing—review and editing; Chao Zhao, Conceptualization, Formal analysis, Supervision, Investigation, Visualization, Methodology; Andrés Cruz-Herranz, Conceptualization, Formal analysis, Investigation, Methodology, Writing—review and editing; Lucile Ben Haim, Christian Cordano, Formal analysis, Investigation, Methodology, Writing—review and editing; Lawrence R Shiow, Conceptualization, Formal analysis, Supervision, Funding acquisition, Methodology, Project administration, Writing—review and editing; Kevin W Kelley, Conceptualization, Investigation, Methodology; Boguslawa Sadowski, Formal analysis, Investigation, Methodology; Garrett Timmons, Jackie N Wright, Jung Hyung Sin, Michael Devereux, Sandra M Chang, Investigation, Methodology; Anne-Katrin Pröbstel, Khalida Sabeur, Investigation, Methodology, Writing—review and editing; Daniel E Morrison, Investigation, Visualization, Methodology; Ari J Green, Klaus-Armin Nave, Robin JM Franklin, Conceptualization, Resources, Supervision, Funding acquisition, Project administration, Writing—review and editing; David H Rowitch, Conceptualization, Resources, Data curation, Supervision, Funding acquisition, Writing—original draft, Project administration, Writing—review and editing

## Author ORCIDs

Lucas Schirmer http://orcid.org/0000-0001-7142-4116
Wiebke Möbius http://orcid.org/0000-0002-2902-7165
Chao Zhao http://orcid.org/0000-0003-1144-1621
Klaus-Armin Nave http://orcid.org/0000-0001-8724-9666
David H Rowitch http://orcid.org/0000-0002-0079-0060

## Ethics

Animal experimentation: All mouse strains were maintained at the University of California, San Francisco (UCSF) specific pathogen-free animal facility under protocol number AN110094. All animal protocols were approved by and in accordance with the guidelines established by the Institutional Animal Care and Use Committee and Laboratory Animal Resource Center.

## Decision letter and Author response

Decision letter https://doi.org/10.7554/eLife.36428.019
Author response https://doi.org/10.7554/eLife.36428.020

# Additional files

## Supplementary files

• Transparent reporting form
DOI: https://doi.org/10.7554/eLife.36428.017

## Data availability

All data generated or analyzed during this study are included in the manuscript and supporting files.

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
