## [Decision Letter]

[Editors’ note: this article was originally rejected after discussions between the reviewers, but the authors were invited to resubmit after an appeal against the decision.]

Thank you for submitting your work entitled "Oligodendrocyte-*Kir4.1* maintains axon integrity and is a therapeutic target in multiple sclerosis" for consideration by *eLife*. Your article has been reviewed by Gary Westbrook as Senior Editor, a Reviewing Editor, and three reviewers. The reviewers have opted to remain anonymous.

Summary:

Our decision has been reached after consultation between the reviewers. In the discussion of the manuscript, all three reviewers had concerns about over-interpretation, particularly with the EAE component, as well as the claims concerning therapeutic implications. The consensus from the discussion of the reviewers was that the work on disease modeling work should be dropped and the manuscript focused on the interesting analysis of the cKOs by presenting the study with more clarity, removing the speculative cartoons and rescue experiments and analyzing the results with appropriate statistical approaches (see specific comments below). Of note, the impact of an oligodendrocyte-specific knockout of Kir4.1 on the integrity of myelin is likely to be small is a concern that is further supported by the recent study of Bergles et al., 2018 observing no differences in myelination in the corpus callosum in oligodendrocyte-specific Kir4.1cKO.

We expect that the changes to your manuscript would require substantial restructuring, and thus we are not able to consider the manuscript in its current form. However, if you decide to address these general issues as above, and the relevant specific comments in the original reviews below, we would be glad to consider a revised and resubmitted manuscript.

Reviewer #1:

This is an important and very timely study building further on previous studies of some of the same authors (Schirmer et al., 2014, Srivastava et al., 2012) in which Kir4.1 autoantibodies were identified in multiple sclerosis (MS) patients. Here, the authors provide an extensive in vivo, ultrastructural and immunofluorescence study of Kir4.1 in the optic nerve (ON), both in progressive MS and in a mouse model of demyelination using conditional Kir4.1 knockouts. They reveal Kir4.1 expression both in the oligodendrocyte cell body and in the myelin sheath based on high-pressure freeze immuno-EM, which is new and exciting. They conclude that long-term impairments of OL Kir4.1 mediate axonal damage and myelin degradation. This part of the work is carefully performed and illustrated by (sometimes excessive) quantification but the second part has major problems.

1) The claim that Kir4.1 represents a therapeutic target (as the title mentions) is not supported by the experiments shown from subsection “Retigabine-induced activation of Kv7.2 channels protects against neurodegeneration in aging *Kir4.1* cKO animals” and onwards. Retigabine (Ezogabine in the US) is not targeting the Kir4.1 channel and neither is it demonstrated to have a convergent action at the glial and neuronal K^+^ gradients as the authors postulate in their cartoons (Figure 2A and Figure 5A). On the contrary, ezogabine is likely to cause a greater extracellular K^+^ accumulation outside of the demyelinated axons and within or around intact internodes and nodes of Ranvier. There is no opening of an additional gate for K^+^ to mitigate the lack of Kir4.1 activity in the cKO mice. There are more errors. For example, there is no evidence that ezogabine shunts the nodal action potential waveform (e.g. Schwartz et al., 2006). A more likely explanation for the partial neuroprotective action of ezogabine is the concomitant reduction in hyperexcitability. Similar protective effects on axon integrity have been observed with the sodium channel blocker phenytoin (Black et al., 2006). Such methodologies are, however, not providing evidence that Kir4.1 is an effective therapeutic target.

2) If the authors wish to demonstrate that Kir4.1 is a specific "therapeutic target" they will need to develop and test the rescue potential of Kir4.1 openers. In the absence of these, my current recommendation is to remove the entire ezogabine part from the manuscript and focus on the in vivo and in vitro findings in the conditional Kir4.1 knockout model. This is a huge amount of high quality data. On the other hand, if the aim is to show the neuroprotective action of ezogabine in MS and propose a repurposing of the drug, there is the need to show immunofluorescence of Kv7.2 and Kv7.3 expression in the ON in demyelinating neuropathies or MS models

3) It remains unclear to me which subcellular sites contribute to the neuropathology in cKO of Kir4.1. That OL cell bodies with Kir4.1 channels are localized to perinodal domains (Figure 2F-G) is not clear from the data but the assumptions widely propagate throughout the manuscript, mainly in cartoon form. But how is the distribution along the myelin sheath different from random and what are the data to support such claim? What are the distinct contributions of cell body Kir4.1 and myelin sheath Kir4.1 to the neuropathology?

Reviewer #2:

Overall this an excellent study by Schirmer and colleagues that contains an impressive amount of convincing experiments and state of the art analyses to comprehensively characterize the role of oligodendroglial Kir4.1 during development, adulthood and in demyelinating diseases.

My only major concerns are with the EAE experiment shown in Figure 6 and they relate to the following two aspects:

1) The clinical course shown in Figure 1B shows a very mild disease course with an average maximal score of 1 in Sal treated mice (corresponding to a limp tail). As the authors also state that for the Sal-RTG group treatment was only started when an EAE score of 2 was reached – this seems nearly incompatible with the data shown here (max group score of 0.5, assuming that the scale is correct). I would suggest to separately analyse the effect on disease incidence (very unlikely to be influenced by a neuroprotective interventions) and the effect on the disease course of those mice that actually develop clinical signs. To show a protective effect of the Sal-RTG vs. Sal-Sal treatment it would e.g. be important to plot those presumably matched mice treated with either Sal-RTG or Sal-Sal aligned to the start of treatment (so the time when these mice have first reached a score of 2).

2) The argument for a primary neuroprotective effect is that neurons and/or axons are preserved, while no effect on the immune response is observed. This argument is difficult to make based on the data presented here – at least the Sal-RTG group does not appear to show histological protection of axons or RGCs. Such protection is observed in the RTG only group however only one immune parameters is analysed here (microglia number is not significantly changed in 5 mice examined – given that the effect size of RTG on microglia number is similar to the effect size on axons and neurons this could also be the result of a the substantially higher variability in this outcome parameter).

In summary to me the claim that RTG is neuroprotective in EAE is the one claim in the manuscript that is currently not sufficiently supported by the data presented here. As this is an important claim for the entire manuscript I would suggest to increase the number of animals in particular in the Sal-RTG vs Sal-Sal group (so treatment only started at an EAE score of 2 with randomization to either RTG or Sal) and analyse both the clinical score as well as the axonal pathology and parameters of adaptive and innate immune cell infiltration in the lesions in a larger number of mice.

Reviewer #3:

In this manuscript, Schirmer et al., explore the role of Kir4.1, a disease-linked inward rectifying potassium channel in oligodendrocytes (OLs), both in the maintenance of axonal integrity and as a potential therapeutic target for multiple sclerosis. The authors find that conditional deletion of Kir4.1 in OLs or OL precursor cells leads to relatively modest defects in OL differentiation and myelination during development. However, as mice age, OL-specific Kir4.1 loss leads to reduced optic nerve function, fewer RGC neurons, some mitochondrial structural abnormalities, motor defects, and early death, supporting an OL-intrinsic function in maintaining axonal integrity. Finally, the authors provide evidence that administration of retigabine (RTG, to open axonal Kv7.2 channels and presumably help buffer extracellular potassium) ameliorates some of the negative effects of Kir4.1 loss in knockout mice or in the EAE mouse model of multiple sclerosis.

Overall the paper is technically rigorous and gives definitive proof that OL-intrinsic Kir4.1 expression is important for function and survival of axons in adult mice, rather than for myelination per se. The weaker part of the story is the disease modeling aspect which is more preliminary and would be improved by additional analysis. Specific points:

1) Throughout the paper, the authors should have performed their statistical analyses using the average measurement for each animal used in the study, rather than on every individual measurement taken. For example, in Figure 2K, statistical analyses on g-ratio seem to have been performed comparing hundreds of axons measured, artificially increasing statistical significance. Same is true for mitochondrial and axonal morphology measurements. Since effect sizes are often very small, it is likely that there is no real statistical or biologically relevant difference between many of these conditions that the authors consider significant.

2) In Figure 1 the authors show that Kir4.1 is reduced in demyelinating lesions. Since Kir4.1 is expressed by mature OLs, does this just reflect loss of mature OLs rather than a specific downregulation of Kir4.1? The plot of MOG (mature OLG marker) versus Kir4.1 in Figure 1E is consistent with this idea.

3) Figure 4: Lysolecithin experiments. The authors conclude that "Kir4.1 is required for white matter integrity after injury" but the data do not robustly support this conclusion. Kir4.1 knockouts have mildly *better* remyelination, consistent with the accelerated differentiation/myelination seen developmentally. There is no data presented on axon degeneration for these experiments, only plots of axon and mitochondrial morphology, all of which suffer from very small effect sizes and inappropriate statistics to suggest these effects are significant (see above point about statistical analysis). The authors state that there is evidence for Wallerian degeneration in Figure 4G but this is just a single TEM micrograph with no quantitative support of this conclusion. I think the paper would be stronger without this set of experiments.

4) In Figure 5 (text in subsection “Retigabine-induced activation of Kv7.2 channels protects against neurodegeneration in aging Kir4.1 cKO animals”), the authors conclude that "RTG-induced activation of Kv7.2 channels protects against neurodegeneration in aging Kir4.1 cKO animals." Therefore, the essential statistical comparison to make is between the groups of mice that were knocked out for Kir 4.1 that either received saline or RTG. The authors do not make this key comparison in their statistical analysis that would support their conclusion, but instead offer statistics for less relevant comparisons (e.g. showing that CKOs treated with saline are statistically different than WT mice treated with RTG is not relevant to the authors' conclusion.)

[Editors’ note: what now follows is the decision letter after the authors submitted for further consideration.]

Thank you for resubmitting your work entitled "Oligodendrocyte-encoded Kir4.1 function is required for axon integrity and long-term maintenance" for further consideration at *eLife*. Your revised article has been favorably evaluated by Gary Westbrook (Senior Editor), and three reviewers.

The authors have substantially refocused the scope of the manuscript and improved its clarity. They investigated myelin integrity and axonal function in the optic nerve in OL-specific Kir4.1 KO during aging and remyelination. The reviewers were all supportive of the work, but we think that several main points require further clarification or discussion. In some cases, this might require additional data and/or toning down of the conclusions.

Essential revisions:

1) One of the new findings is the localization of Kir4.1 within the myelin sheath at the membrane. Although such location was speculated previously (e.g. in Larson et al., 2018) published work so far has only detect Kir4.1 in OL cell bodies and processes (Brasko et al., 2017; Kalsi et al., 2004; Poopalasundaram et al., 2000; Battefeld et al., 2016; Schirmer et al., 2014). Thus, it is important that the authors provide further evidence confirming the specificity of the immunoEM labeling. The optimal control is to stain the conditional knockouts; second best is a no-primary control to look at how "sticky" the tissue is. How many EM sections were analyzed. How do the data compare to background? Were the *Olig2-cre:Kir4.1* or *Cnp-cre:Kir4.1* mice examined with immuno EM to assess the specificity of the labeling?

In addition, the scale bars seem to be wrong. Gold particles are ~10 nm which is not consistent with the scale bars in Figure 1F.

2) The impact on mitochondria integrity is assessed on basis of their numbers and circularity. However, higher magnification images showing mitochondria are needed to assess the validity of these differences between the groups.

3) The authors provide minimal context how their manuscript relates to the previously published study by Larson et al., (2018). In the Introduction they write "we focused on late roles of OL-Kir4.1 in mature adults." The scientific or conceptual advance relative to the published work needs to be honestly and forthrightly discussed.

---

## [Author Response]

[Editors’ note: the author responses to the first round of peer review follow.]

Reviewer #1:This is an important and very timely study building further on previous studies of some of the same authors (Schirmer et al., 2014, Srivastava et al., 2012) in which Kir4.1 autoantibodies were identified in multiple sclerosis (MS) patients. Here, the authors provide an extensive in vivo, ultrastructural and immunofluorescence study of Kir4.1 in the optic nerve (ON), both in progressive MS and in a mouse model of demyelination using conditional Kir4.1 knockouts. They reveal Kir4.1 expression both in the oligodendrocyte cell body and in the myelin sheath based on high-pressure freeze immuno-EM, which is new and exciting. They conclude that long-term impairments of OL Kir4.1 mediate axonal damage and myelin degradation. This part of the work is carefully performed and illustrated by (sometimes excessive) quantification but the second part has major problems.1) The claim that Kir4.1 represents a therapeutic target (as the title mentions) is not supported by the experiments shown from subsection “Retigabine-induced activation of Kv7.2 channels protects against neurodegeneration in aging Kir4.1 cKO animals” and onwards. Retigabine (Ezogabine in the US) is not targeting the Kir4.1 channel and neither is it demonstrated to have a convergent action at the glial and neuronal K^+^ gradients as the authors postulate in their cartoons (Figure 2A and Figure 5A). On the contrary, ezogabine is likely to cause a greater extracellular K^+^ accumulation outside of the demyelinated axons and within or around intact internodes and nodes of Ranvier. There is no opening of an additional gate for K^+^ to mitigate the lack of Kir4.1 activity in the cKO mice. There are more errors. For example, there is no evidence that ezogabine shunts the nodal action potential waveform (e.g. Schwartz et al., 2006). A more likely explanation for the partial neuroprotective action of ezogabine is the concomitant reduction in hyperexcitability. Similar protective effects on axon integrity have been observed with the sodium channel blocker phenytoin (Black et al., 2006). Such methodologies are, however, not providing evidence that Kir4.1 is an effective therapeutic target.

We take the reviewer’s point and based on this suggestion and similar advice from the editor and other reviewers we have removed this data, which will be augmented for an independent study. The suggestions above for additional endpoints and potential mechanisms are helpful and will be pursued in this future work.

2) If the authors wish to demonstrate that Kir4.1 is a specific "therapeutic target" they will need to develop and test the rescue potential of Kir4.1 openers. In the absence of these, my current recommendation is to remove the entire ezogabine part from the manuscript and focus on the in vivo and in vitro findings in the conditional Kir4.1 knockout model. This is a huge amount of high quality data. On the other hand, if the aim is to show the neuroprotective action of ezogabine in MS and propose a repurposing of the drug, there is the need to show immunofluorescence of Kv7.2 and Kv7.3 expression in the ON in demyelinating neuropathies or MS models

As suggested, we removed the data on retigabine (ezogabine) as a pharmacological model to improve function in Kir4.1 cKO animals and focused on characterizing the Kir4.1 knockout model during development/aging and white matter injury.

3) It remains unclear to me which subcellular sites contribute to the neuropathology in cKO of Kir4.1. That OL cell bodies with Kir4.1 channels are localized to perinodal domains (Figure 2F-G) is not clear from the data but the assumptions widely propagate throughout the manuscript, mainly in cartoon form. But how is the distribution along the myelin sheath different from random and what are the data to support such claim? What are the distinct contributions of cell body Kir4.1 and myelin sheath Kir4.1 to the neuropathology?

We found that Kir4.1 channels are localized in two peri-axonal parts of the oligodendrocytes (juxta-axonal and peri-nodal) and we think that these components are best positioned to explain the pathology of axon degeneration in the Kir4.1 cKO animals. However, we cannot rule out that cell body Kir4.1 also has a role (e.g., in regulating OL differentiation during early steps of development) and a caveat has been added to the Discussion section.

Reviewer #2:Overall this an excellent study by Schirmer and colleagues that contains an impressive amount of convincing experiments and state of the art analyses to comprehensively characterize the role of oligodendroglial Kir4.1 during development, adulthood and in demyelinating diseases.My only major concerns are with the EAE experiment shown in Figure 6 and they relate to the following two aspects:1) The clinical course shown in Figure 1B shows a very mild disease course with an average maximal score of 1 in Sal treated mice (corresponding to a limp tail). As the authors also state that for the Sal-RTG group treatment was only started when an EAE score of 2 was reached – this seems nearly incompatible with the data shown here (max group score of 0.5, assuming that the scale is correct). I would suggest to separately analyse the effect on disease incidence (very unlikely to be influenced by a neuroprotective interventions) and the effect on the disease course of those mice that actually develop clinical signs. To show a protective effect of the Sal-RTG vs. Sal-Sal treatment it would e.g. be important to plot those presumably matched mice treated with either Sal-RTG or Sal-Sal aligned to the start of treatment (so the time when these mice have first reached a score of 2).

We thank the reviewer for the comments and suggestions. As indicated in response reviewer 1, point 1, we removed the entire part on EAE and retigabine (ezogabine) rescue trial but will keep this in mind for a future publication and repeat EAE analyses.

2) The argument for a primary neuroprotective effect is that neurons and/or axons are preserved, while no effect on the immune response is observed. This argument is difficult to make based on the data presented here – at least the Sal-RTG group does not appear to show histological protection of axons or RGCs. Such protection is observed in the RTG only group however only one immune parameters is analysed here (microglia number is not significantly changed in 5 mice examined – given that the effect size of RTG on microglia number is similar to the effect size on axons and neurons this could also be the result of a the substantially higher variability in this outcome parameter).

We agree with the reviewer and will include additional EAE cohorts with respect to a preclinical RTG trial in a future paper. Also, we will study different inflammatory parameters such as T cell infiltration of presence of phagocytes. For now, we removed the part on EAE and instead focused entirely on the functions of OL-Kir4.1 during development, adulthood and its novel role in long-term axon support.

In summary to me the claim that RTG is neuroprotective in EAE is the one claim in the manuscript that is currently not sufficiently supported by the data presented here. As this is an important claim for the entire manuscript I would suggest to increase the number of animals in particular in the Sal-RTG vs Sal-Sal group (so treatment only started at an EAE score of 2 with randomization to either RTG or Sal) and analyse both the clinical score as well as the axonal pathology and parameters of adaptive and innate immune cell infiltration in the lesions in a larger number of mice.

We agree with the reviewer that we cannot confidently say that retigabine (ezogabine) treatment has direct neuroprotective effects without more evidence and this data is now removed from the paper.

Reviewer #3:In this manuscript, Schirmer et al., explore the role of Kir4.1, a disease-linked inward rectifying potassium channel in oligodendrocytes (OLs), both in the maintenance of axonal integrity and as a potential therapeutic target for multiple sclerosis. The authors find that conditional deletion of Kir4.1 in OLs or OL precursor cells leads to relatively modest defects in OL differentiation and myelination during development. However, as mice age, OL-specific Kir4.1 loss leads to reduced optic nerve function, fewer RGC neurons, some mitochondrial structural abnormalities, motor defects, and early death, supporting an OL-intrinsic function in maintaining axonal integrity. Finally, the authors provide evidence that administration of retigabine (RTG, to open axonal Kv7.2 channels and presumably help buffer extracellular potassium) ameliorates some of the negative effects of Kir4.1 loss in knockout mice or in the EAE mouse model of multiple sclerosis.Overall the paper is technically rigorous and gives definitive proof that OL-intrinsic Kir4.1 expression is important for function and survival of axons in adult mice, rather than for myelination per se. The weaker part of the story is the disease modeling aspect which is more preliminary and would be improved by additional analysis. Specific points:1) Throughout the paper, the authors should have performed their statistical analyses using the average measurement for each animal used in the study, rather than on every individual measurement taken. For example, in Figure 2K, statistical analyses on g-ratio seem to have been performed comparing hundreds of axons measured, artificially increasing statistical significance. Same is true for mitochondrial and axonal morphology measurements. Since effect sizes are often very small, it is likely that there is no real statistical or biologically relevant difference between many of these conditions that the authors consider significant.

We agree with the reviewer and provided additional statistical analyses using the average measurement for each animal when comparing EM-derived data like g-ratios, axon diameter and mitochondrial parameters (please see revised Figure 2, Figure 4 and Figure 5 and corresponding Supplementary Figures). Also, we included statistics on counts for intra-axonal mitochondria per axon versus mitochondrial densities in situations where axons are pathologically enlarged such as chronic lysolecithin lesions (cf. Figure 5) and cited additional references (Kiryu-Seo et al., 2010 and Zambonin et al., 2011) to emphasize increase in intraaxonal mitochondria densities and stationary mitochondrial size in remyelination and lysolecithin-mediated white matter injury (Discussion section) (Kiryu-Seo et al., 2010; Zambonin et al., 2011).

2) In Figure 1 the authors show that Kir4.1 is reduced in demyelinating lesions. Since Kir4.1 is expressed by mature OLs, does this just reflect loss of mature OLs rather than a specific downregulation of Kir4.1? The plot of MOG (mature OLG marker) versus Kir4.1 in Figure 1E is consistent with this idea.

The reviewer asks whether loss of Kir4.1 occurs because OLs are lost. We think this is not the case in acute lesions where KIr4.1 becomes downregulated. Also, our loss-of-function studies show that Kir4.1 function is not needed to re-myelinate focal white matter lesions. So, in the acute lesions our conclusion is that Kir4.1 downregulation precedes loss of OLs. However, we do agree with the reviewer about the links in chronic demyelinated lesions that leads to a loss of OLs. Note that because OPCs express little Kir4.1 it is formally possible that OPC differentiation is blocked resulting in Kir4.1 lack of expression. As we removed translational aspects on MS and EAE in the current revised manuscript, we decided not to comment on this issue in the paper but will follow-up on the issue in a follow-up study.

3) Figure 4: Lysolecithin experiments. The authors conclude that "Kir4.1 is required for white matter integrity after injury" but the data do not robustly support this conclusion. Kir4.1 knockouts have mildly *better* remyelination, consistent with the accelerated differentiation/myelination seen developmentally. There is no data presented on axon degeneration for these experiments, only plots of axon and mitochondrial morphology, all of which suffer from very small effect sizes and inappropriate statistics to suggest these effects are significant (see above point about statistical analysis). The authors state that there is evidence for Wallerian degeneration in Figure 4G but this is just a single TEM micrograph with no quantitative support of this conclusion. I think the paper would be stronger without this set of experiments.

We take the reviewer’s suggestions and now provide additional statistical analyses using the average measurement for each animal (see above, Figure 5).

We removed the term Wallerian degeneration and now describe pathological axon changes like changes in size and mitochondria, which can be reliably quantified.

We think that the data sets on acute and chronic white matter injury (14dpl and 60dpl) are important for the current paper to indicate the timing of white matter pathology onset observed in acute versus chronic lesions. Indeed, long-term lesions together with functional and histological results from P140 and P180 allows to parse out a specific function of OL-Kir4.1 for long-term but not early support of axon function.

4) In Figure 5 (text in subsection “Retigabine-induced activation of Kv7.2 channels protects against neurodegeneration in aging Kir4.1 cKO animals”), the authors conclude that "RTG-induced activation of Kv7.2 channels protects against neurodegeneration in aging Kir4.1 cKO animals." Therefore, the essential statistical comparison to make is between the groups of mice that were knocked out for Kir 4.1 that either received saline or RTG. The authors do not make this key comparison in their statistical analysis that would support their conclusion, but instead offer statistics for less relevant comparisons (e.g. showing that CKOs treated with saline are statistically different than WT mice treated with RTG is not relevant to the authors' conclusion.)

As above, we removed the entire part on retigabine (ezogabine) as suggested by the editor and other reviewers but will consider those comments and perform additional analyses focusing only on the two cKO groups in a future study.

[Editors’ note: the author responses to the re-review follow.]

The authors have substantially refocused the scope of the manuscript and improved its clarity. They investigated myelin integrity and axonal function in the optic nerve in OL-specific Kir4.1 KO during aging and remyelination. The reviewers were all supportive of the work, but we think that several main points require further clarification or discussion. In some cases, this might require additional data and/or toning down of the conclusions.Essential revisions:1) One of the new findings is the localization of Kir4.1 within the myelin sheath at the membrane. Although such location was speculated previously (e.g. in Larson et al., 2018) published work so far has only detect Kir4.1 in OL cell bodies and processes (Brasko et al., 2017; Kalsi et al., 2004; Poopalasundaram et al., 2000; Battefeld et al., 2016; Schirmer et al., 2014). Thus, it is important that the authors provide further evidence confirming the specificity of the immunoEM labeling. The optimal control is to stain the conditional knockouts; second best is a no-primary control to look at how "sticky" the tissue is. How many EM sections were analyzed. How do the data compare to background? Were the Olig2-cre:Kir4.1 or Cnp-cre:Kir4.1 mice examined with immuno EM to assess the specificity of the labeling?

We thank the reviewer for this important question and have included an in-depth analysis to confirm specificity of the Kir4.1 immunogold labeling with respect to myelin and astrocyte compartments. We performed a no-primary control and confirmed specificity of the Kir4.1 antibody labeling with the strongest difference at the inner myelin tongue highlighting the potential functional importance of this particular localization. Furthermore, we analyzed immunogold labeling in both *Olig2*- and *Cnp-cre Kcnj10* cKO optic nerve tissues as compared to controls and found a specific lack of labeling in myelin compartments, in particular at the inner tongue and within the compact myelin. Outer tongue and astrocyte labeling was not affected by the cKO; of note, outer tongue labeling might belong to astrocytes due to the direct attachment of astrocyte fibers to myelin; text in the main manuscript (subsection “OL-*Kir4.1* channels are gradually upregulated during early postnatal development and show a peri-axonal expression pattern”), figures and corresponding legends have been revised accordingly

In addition, the scale bars seem to be wrong. Gold particles are ~10 nm which is not consistent with the scale bars in Figure 1F.

We thank the reviewer for this comment and have modified the scale bars and revised the figures accordingly.

2) The impact on mitochondria integrity is assessed on basis of their numbers and circularity. However, higher magnification images showing mitochondria are needed to assess the validity of these differences between the groups.

We agree with the reviewer’s comment and have included high-magnification images of representative intra-axonal mitochondria from control and cKO lesioned spinal cord tissue (short- and long-term remyelination) in Figure 5; figures and corresponding legends have been revised accordingly.

3) The authors provide minimal context how their manuscript relates to the previously published study by Larson et al., (2018). In the Introduction they write "we focused on late roles of OL-Kir4.1 in mature adults." The scientific or conceptual advance relative to the published work needs to be honestly and forthrightly discussed.

Our study is novel in showing the role of oligodendrocyte Kir4.1 in in long-term maintenance of axon function and integrity in long white matter tracts, such as the optic nerves and the spinal white matter. We think that both studies are complementary showing diverse roles for OL-Kir4.1 in regulating neuroaxonal excitability and function acutely and over a longer time course during white matter injury and aging. We modified the text in the Introduction and Discussion section to clarify these points.